# Dropout Universality: Scaling Laws and Optimal Scheduling at the Edge-of-Chaos

**Lucas Fernandez Sarmiento** [1]

## Abstract

We develop a mean-field theory of dropout as a perturbation of critical signal propagation at the edge of chaos, and show that it predicts a simple, no-cost change to standard practice: *front-loaded* dropout schedules cut test loss by 18–35% over constant dropout in MLPs and Vision Transformers at fixed budget. The theoretical mechanism is that dropout shifts the perfect-alignment fixed point, making the depth scale for information propagation finite even at critical initialization. We derive critical and crossover scaling laws for correlation decay and establish that smooth activations and kinked, $\mathrm{ReLU}$-like activations constitute distinct universality classes, with different critical exponents and a universal two-parameter scaling collapse in detuning and dropout strength. The distinction traces to the analytic structure of the correlation map: smooth activations admit a Taylor expansion near perfect alignment, while kinked activations develop a branch point with universal non-analyticity. As a corollary, the framework yields saturated dropout profiles under fixed budget; a regularization-reach argument then selects front-loaded schedules, with accuracy gains as a consistent secondary effect. We also discuss how the same Gaussian-kernel structure extends the theory beyond MLPs toward CNNs and residual architectures.

## 1. Introduction

Mean-field analyses of randomly initialized deep networks reveal an order-to-chaos phase diagram that controls both signal propagation and gradient penetration with depth (Poole et al., 2016; Schoenholz et al., 2017; Bahri et al.,

2020; 2024b). In this work, we study how this picture is modified by dropout, and use the resulting theory to derive a no-cost change to standard practice: we find that front-loaded dropout schedules cut test loss by 18–35% over constant dropout at fixed budget in MLPs and Vision Transformers (Table 3). Using the representation-group (coarse-graining) language of (Roberts et al., 2022; Bahri et al., 2024b), we show that dropout behaves as a relevant perturbation: it displaces the critical fixed point and grows under depth coarse-graining, pushing the dynamics away from criticality and determining the macroscopic phase. Concretely, dropout deforms the correlation map (Schoenholz et al., 2017) by adding a correlation-independent shift at perfect alignment, so that perfect correlation between inputs is no longer a fixed point for any nonzero dropout. Thus, even at the edge-of-chaos (Sompolinsky et al., 1988; Packard, 1988; Bertschinger & Natschläger, 2004), the depth correlation length is finite. We interpret this shift using Landau theory, as an equation of state for the decorrelation order parameter $m \equiv 1 - c^*$ (with $c^*$ the asymptotic inter-signal correlation), derive the resulting scaling laws, and show that smooth and kinked, $\mathrm{ReLU}$-like activations exhibit the same qualitative phase diagram but different critical scaling in the presence of dropout.

Additionally, we show that the two control parameters (dropout strength and distance to criticality) admit a scaling collapse into a single universal form,[1] and we illustrate how these predictions compare with mean-field recursions.

The main conceptual contributions are threefold: first, while dropout was previously shown to destroy the order-to-chaos critical point (Schoenholz et al., 2017), we show that the deformed mean-field map still retains a nontrivial $c^* < 1$ fixed point. This allows us to have a well-defined correlation length recursion even in the presence of dropout, and yields a Landau equation of state for $m = 1 - c^*$ together with a two-parameter scaling collapse. Second, we show that universality classes and scaling laws are set by the analytic structure of activations: smooth and kinked activations

---

[1]Carnegie Mellon University (CMU), Pittsburgh, Pennsylvania, USA. Correspondence to: Lucas Fernandez Sarmiento <lucasf@andrew.cmu.edu>.

*Proceedings of the $43^{rd}$ International Conference on Machine Learning*, Seoul, South Korea. PMLR 306, 2026. Copyright 2026 by the author(s).

---

[1]This is analogous to the homogeneous scaling of connected correlation functions near a critical point; related scaling forms hold for one-particle-irreducible vertex functions after the usual Legendre transform (Zinn-Justin, 2002).

have distinct critical exponents (Table 2), a distinction reflected in their Hermite spectral structure (App. C). Since the same Gaussian activation kernels arise beyond MLPs, including CNNs and residual branches (App. A.4), we expect the qualitative smooth/kinked split to be a well-motivated heuristic. Third, we treat dropout as a depth-dependent dynamical field; treating classically fixed hyperparameters as fields that can change through the network opens a new dimension for hyperparameter optimization across depth, and potentially over training time.

This work complements prior studies of depth-dependent regularization: stochastic depth drops whole residual blocks with increasing probability (Huang et al., 2016), curriculum dropout anneals dropout over training time (Morerio et al., 2017), and LayerDrop removes transformer layers for efficiency (Fan et al., 2020). Our schedules are spatial depth profiles, so they are complementary to temporal curricula and layer-dropping mechanisms. Qualitatively different edge-of-chaos behavior for smooth and $\mathrm{ReLU}$-like activations was previously observed in (Hayou et al., 2019); here we extract the scaling exponents and identify the corresponding universality classes.[2]

**Conflict of Interest Disclosure.** The author declares no financial conflicts of interest related to this work.

## 2. Mean-Field Theory Background

We first state the mean-field assumptions for MLPs, which provide the baseline for the dropout analysis (Poole et al., 2016; Schoenholz et al., 2017; Bahri et al., 2020; 2024b). We consider fully-connected MLPs at random initialization, where preactivations are well-approximated as Gaussian random variables with self-consistently determined statistics (means, variances, covariances). In the strict infinite width limit this Gaussian description becomes exact (Lee et al., 2018), while at large but finite width deviations from Gaussianity can be organized perturbatively in the depth-to-width aspect ratio (schematically $L/N$), as treated comprehensively in (Roberts et al., 2022; Bahri et al., 2024b). The intuition behind this treatment is that as we add more activations, through the CLT, the statistics will become more and more Gaussian, approaching this distribution asymptotically in the large N limit.

We consider an MLP of depth $L$, $N_l$ neurons in layer $l$, and an activation $\phi : \mathbb{R} \to \mathbb{R}$ (e.g. $\tanh$, ReLU, *etc.*). We use the standard i.i.d. Gaussian initialization

$$W_{ij}^l \sim \mathcal{N}\left(0, \frac{\sigma_w^2}{N_l}\right), \qquad b_i^l \sim \mathcal{N}\left(0, \sigma_b^2\right), \quad (1)$$

---

[2]Code, configurations, seeds, and figure-generation scripts are available in the commit-pinned `dropout-universality-experiments` repository. See App. D for the full reproducibility statement.

take the input as $y_{i;a}^0 = x_{i;a}$, and propagate forward via

$$z_{i;a}^l = W_{ij}^l y_{j;a}^l + b_i^l, \qquad y_{i;a}^{l+1} = \phi\left(z_{i;a}^l\right), \quad (2)$$

where $a$ labels the input. Throughout, expectations $\mathbb{E}[\cdot]$ are over the random weights and biases at initialization, with inputs held fixed. We also use the standard Gaussian measure

$$\int Dz(\cdots) \equiv \frac{1}{\sqrt{2\pi}} \int_{-\infty}^{\infty} dz e^{-z^2/2}(\cdots), \quad (3)$$

and similarly for $\int Dz_1 Dz_2$.

In the infinite-width limit this becomes exact layer-by-layer, while at finite width the first layer is exactly Gaussian (being a linear map of fixed inputs) and subsequent layers are only approximately Gaussian, with controlled non-Gaussian corrections parametrized by the depth-to-width ratio (Roberts et al., 2022). A fuller treatment of the MFT background is given in App. A.2.

We track the single-input preactivation variance $q_{aa}^l$, the two-input covariance $q_{ab}^l$, and the induced correlation $c_{ab}^l$:

$$q_{aa}^l = \sigma_w^2 \int Dz \phi^2\left(\sqrt{q_{aa}^{l-1}} z\right) + \sigma_b^2 \quad (4)$$

$$q_{ab}^l = \sigma_w^2 \int Dz_1 Dz_2 \phi(u_1)\phi(u_2) + \sigma_b^2 \quad (5)$$

$$u_1 = \sqrt{q_{aa}^{l-1}} z_1, \qquad c_{ab}^{l-1} = \frac{q_{ab}^{l-1}}{\sqrt{q_{aa}^{l-1} q_{bb}^{l-1}}}, \quad (6)$$

$$u_2 = \sqrt{q_{bb}^{l-1}}\left(c_{ab}^{l-1} z_1 + \sqrt{1 - \left(c_{ab}^{l-1}\right)^2} z_2\right). \quad (7)$$

In the absence of dropout, for a wide array of activations $q_{ab}^l, c_{ab}^l$ settle to fixed points $q^*, c^*$; in particular, there exists a $c^* = 1$ fixed point as a straightforward calculation shows. The variance fixed point sets the typical activation scale, while the correlation fixed point describes whether distinct inputs become indistinguishable under depth iteration. We denote the recurrence relation for $c_{ab}^l$ by $c_{ab}^l = F(c_{ab}^{l-1})$.

To probe its stability, one linearizes the map at $c = 1$, defining the (angular) susceptibility

$$\chi_\perp \equiv \left.\frac{\partial c_{ab}^l}{\partial c_{ab}^{l-1}}\right|_{c_{ab}=1} = \sigma_w^2 \int Dz\left[\phi'(\sqrt{q^*}z)\right]^2. \quad (8)$$

This creates a phase diagram with three regimes: $\chi_\perp < 1$ is the ordered regime, $\chi_\perp > 1$ is chaotic, and $\chi_\perp = 1$ defines the critical regime (the so-called "edge-of-chaos"). The depth scale controlling cross-input correlations can become arbitrarily large in the latter, allowing information to penetrate deep into the network. For ReLU, the criticality condition gives $\sigma_w^2 = 2$, coinciding with the variance used

in He initialization (He et al., 2015)[3] (see App. A.2), a more fundamental viewpoint than variance preservation.

This leads to an emergent characteristic depth scale $\xi_c$, which parametrizes both signal propagation and gradient flow. Iterating random affine maps induces an effective coarse-graining in depth akin to renormalization-group flow, called representation group flow in the ML context (Roberts et al., 2022); information about norms and angles relaxes exponentially fast,

$$|c^* - c^l_{ab}| \propto e^{-l/\xi_c}, \qquad \xi_c^{-1} \equiv -\log|r_c|. \qquad (9)$$

An analogous correlation length $\xi_q$ controls how quickly single-input norms settle to $q^*$. Heuristically, $\xi_c$ controls how far distinctions between different inputs survive with depth; in the ordered phase $c^* = 1$, so $u_1^* = u_2^*$ and $\xi_c^{-1} = -\log\chi_\perp$, hence $\chi_\perp \to 1$ implies $\xi_c \to \infty$. Consequently, the edge-of-chaos is primarily about $\xi_c$ rather than $\xi_q$, except in special cases. For $\mathrm{ReLU}$, the curvature vanishes everywhere away from the origin, leading to a coincidence between $\xi_c$ and $\xi_q$, both diverging as $\chi_\perp \to 1$, provided a finite $q^*$ exists. This is one concrete reason why $\mathrm{ReLU}$-style near-critical initializations are comparatively forgiving in practice.

As previously discussed, mean-field theory is an expansion in non-Gaussian corrections suppressed by powers of $L/N$. Therefore, in our experiments we work in a controlled regime with $L/N \ll 1$ (i.e., $N \gg L$), where the large-width expansion is perturbatively sound (Roberts et al., 2022). The correlation length diverges as we approach criticality, amplifying finite-width and other non-Gaussian corrections. Thus, when probing critical power laws with dropout, we keep the dropout rate small but non-negligible: infinitesimal dropout would require prohibitively deep networks to estimate the correlation length and related quantities.

Although we derive these results for MLPs, analogous large-width limits exist elsewhere. CNNs are the canonical convolutional setting (LeCun et al., 1998), and at infinite channel count their covariance recursions again close through Gaussian activation kernels (Xiao et al., 2018). ResNets are especially suggestive: Yang and Schoenholz (Yang & Schoenholz, 2017) find qualitatively different tanh/$\mathrm{ReLU}$ large-depth behavior. Skip connections alter global depth dynamics, easing the information propagation by injecting the original signal after every layer, changing exponential convergence to subexponential or polynomial laws and allowing norm drift, but each residual branch still inherits the local nonlinear Gaussian kernel. Thus our smooth/kinked universality split offers a possible theoretical explanation for their observed dichotomy; a dropout-deformed ResNet theory is left for future work. For transformers (Vaswani

et al., 2017), related infinite-width attention analyses use large hidden dimension and/or many heads (Hron et al., 2020). Because the exponents are controlled by the local analytic structure of this channel, we expect the mechanism to persist when it controls the wide-limit recursion. In Sec. 3.3, we test this extrapolation in transformers.

In App. D we probe realistic scenarios across architectures, datasets, and schedules. We focus on overparameterized regimes where dropout improves generalization and accuracy, and thus where it is used in practice, and we explicitly study the trade-off between dropout-driven regularization and stable signal propagation. The mean-field objective predicts sparse, step-like allocation at fixed budget, but it is permutation invariant in depth and therefore cannot by itself decide where dropout should be concentrated. To break this degeneracy, we use the same correlation length to measure regularization reach: early masks perturb a longer downstream suffix before their effect is absorbed, making the first layers the most valuable place to inject noise.

## 3. Universality Classes and Critical Scaling under Dropout

Dropout (Srivastava et al., 2014; Wager et al., 2013) is often used as a regularizer that reduces overfitting by randomly masking activations during training, thereby reducing the network's reliance on any single subset of pathways and discouraging co-adaptation. In the previous literature, dropout was explored in MFT (Huang et al., 2020; Schoenholz et al., 2017), but not in the universality class sense. We formalize dropout as multiplicative Bernoulli noise applied post-activation,

$$p_j^{(\ell)} \sim \mathrm{Bernoulli}(\rho), \qquad y_j^{(\ell)} \mapsto \tilde{y}_j^{(\ell)} \equiv \frac{p_j^{(\ell)}}{\rho}y_j^{(\ell)}, \quad (10)$$

i.e. the standard inverted-dropout convention in which $\mathbb{E}_p[\tilde{y}_j^{(\ell)}] = y_j^{(\ell)}$, and hence $\mathbb{E}_p[\tilde{z}_i^\ell \mid W, b] = z_i^\ell$, so the mean preactivation is invariant to the dropout rate (Schoenholz et al., 2017). Throughout, $\rho$ is the keep probability. When comparing two inputs, we draw the dropout masks independently for each input[4]; this independence is the source of the decorrelation effect analyzed below.

This distinction originally emerged from the following variational question: among activations of fixed scale, which functions make the correlation length of the near-critical Gaussian channel as large as possible? Extremizing the correlation length functional shows that the natural eigen-

---

[3]This is not true for general activations, but rather an idiosyncrasy of ReLU.

[4]Correlated dropout masks can alleviate this detuning: if the same mask is shared across the batch, the $c = 1$ fixed point is preserved and near-critical scaling can be recovered by retuning, but the noise becomes batch coupled and the regularization is correspondingly weaker and more structured. We leave detailed analysis of correlated masks to future work.

functions of this problem are Hermite polynomials. When standard nonlinearities are decomposed in that basis, two qualitatively different spectra emerge. Smooth, analytic activations such as $\tanh$ have exponentially decaying Hermite coefficients, so their effect is dominated by the first few modes. Kinked activations such as ReLU have power-law tails instead: the kink keeps exciting higher Hermite modes, leaving a broader spectrum in the correlation map. This spectral split was the original diagnostic for the smooth/kinked universality classes, which later reappears in the dropout scaling laws. See App. C for detailed calculations.

Our classification is orthogonal to the scale-invariant vs. non-scale-invariant distinction of (Roberts et al., 2022): the multi-scale Hermite spectrum highlighted above is controlled primarily by analytic structure, not by scale invariance per se. The criterion is based on two facts: (a) near-critical scaling laws differ between smooth and kinked activations, and (b) as previewed above, the Hermite spectral weight of non-analytic activations decays as a power law, in contrast to the exponential decay seen in smooth activations (Apps. B and C).

Extending the aforementioned mean-field framework to include dropout, one finds that for a single input dropout largely acts as a renormalization of the effective weight variance, $\sigma_w^2 \mapsto \sigma_w^2/\rho$, but for two inputs the perfect-alignment fixed point at $c^* = 1$ is lost for any $\rho < 1$ when the two dropout masks are chosen independently (Schoenholz et al., 2017). In other words, dropout acts as an explicit symmetry-breaking "field" for alignment: even if two inputs are perfectly aligned at some depth, the next layer produces $\bar{c}_{ab} < 1$, cutting off the divergence of the depth correlation length at the edge.

Introducing multiplicative Bernoulli noise with keep probability $\rho$, the perturbed preactivations read

$$z_i^\ell = \frac{1}{\rho} \sum_j W_{ij}^\ell p_j^\ell y_j^\ell + b_i^\ell. \qquad (11)$$

The single-input variance recursion becomes

$$\bar{q}_{aa}^\ell = \frac{\sigma_w^2}{\rho} \int Dz \phi^2 \left( \sqrt{\bar{q}_{aa}^{\ell-1}} z \right) + \sigma_b^2, \qquad (12)$$

where the bar reminds us that the variance fixed point is shifted by dropout. We denote the (putative) dropout variance fixed point by

$$\bar{q}^* = \frac{\sigma_w^2}{\rho} \int Dz \phi^2 (\sqrt{\bar{q}^*} z) + \sigma_b^2. \qquad (13)$$

For two distinct inputs $a \neq b$, we find a dropout shift at perfect alignment. If at some depth $c_{ab}^\ell = 1$, then at the next layer one finds (Schoenholz et al., 2017)

$$\bar{F}_\rho(1) = \bar{c}_{ab}^{\ell+1} = 1 - \frac{1-\rho}{\rho \bar{q}^*} \sigma_w^2 \int Dz \phi^2 (\sqrt{\bar{q}^*} z). \qquad (14)$$

It is therefore natural to define the dropout field

$$\begin{aligned} h &\equiv 1 - \bar{F}_\rho(1) \\ &= \frac{1-\rho}{\rho \bar{q}^*} \sigma_w^2 \int Dz \phi^2 (\sqrt{\bar{q}^*} z). \end{aligned} \qquad (15)$$

By construction, $\rho < 1 \Rightarrow h > 0$, so the fixed point at $c^* = 1$ is immediately spoiled: even if two inputs are perfectly aligned at some depth, the next layer produces $\bar{c}_{ab} = 1 - h < 1$. Since we focus on weak dropout, it is useful to perform the small-$\delta\rho$ expansion. Writing $\delta\rho \equiv 1 - \rho$ with $\delta\rho \ll 1$, we have $\bar{q}^* = q^* + \mathcal{O}(\delta\rho)$, so to leading order we have

$$h = \delta\rho \frac{\sigma_w^2}{\bar{q}^*} \int Dz \phi^2 (\sqrt{\bar{q}^*} z) + \mathcal{O}(\delta\rho^2). \qquad (16)$$

The backward pass has the same source of asymmetry. Let $\delta_{j,a}^\ell$ be the backpropagated preactivation gradient for input $a$. With inverted dropout,

$$\delta_{j,a}^\ell = \frac{p_{j,a}^\ell}{\rho} \phi'(u_{j,a}^\ell) \sum_i W_{ij}^{\ell+1} \delta_{i,a}^{\ell+1}, \qquad (17)$$

up to convention-dependent layer indexing. If $G_{ab}^\ell \equiv \mathbb{E}[\delta_a^\ell \delta_b^\ell]$ denotes the two-input gradient covariance, the wide-limit recursion contains

$$G_{ab}^\ell = \sigma_w^2 \mathbb{E}\left[ \frac{p_a^\ell p_b^\ell}{\rho^2} \phi'(u_a) \phi'(u_b) \right] G_{ab}^{\ell+1}. \qquad (18)$$

Thus independent masks give a diagonal factor $\mathbb{E}[p_a^2]/\rho^2 = 1/\rho$ but an off-diagonal factor $\mathbb{E}[p_a p_b]/\rho^2 = 1$. Dropout inflates single-sample gradient variance without similarly inflating cross-sample covariance, mirroring the forward asymmetry that produced $h$. After normalization this suggests a backward field $h_\nabla(\rho)$ that moves perfect gradient alignment below one. We do not identify it with the forward field: the backward kernel is built from $\phi'$, not $\phi$, and for ReLU $\phi'$ is discontinuous. A full dropout-deformed backward critical theory, including finite-width gradient susceptibilities and training-time mask correlations, is left for future work.

### 3.1. Landau Theory and Critical Exponents

For smooth activations, the dropout-deformed correlation edge admits a simple Landau-style description in terms of three mean-field variables:

$$m \equiv 1 - c^*, \qquad t \equiv \chi_\rho - 1, \qquad h \equiv 1 - \bar{F}_\rho(1). \quad (19)$$

The fields $(m, t, h)$ are chosen to mirror statistical-mechanics notation: $t$ plays the role of a reduced temperature, $m$ is the order parameter (magnetization), and $h$ is an external field conjugate to $m$ that explicitly biases the system away from perfect alignment.

We assume inverted dropout with keep probability $\rho \in (0, 1]$, with masks drawn i.i.d. across units and independently across inputs. Let $\bar{q}^*$ denote the dropout variance fixed point (13). For an activation for which Price's theorem applies and $\phi', \phi'' \in L^2(\mathcal{N}(0, \bar{q}^*))$ (for instance, $\phi \in C^2$ with sufficient Gaussian integrability) repeated application of Price's theorem gives the first two derivatives of the correlation map at perfect alignment,[5]

$$\chi_\rho \equiv \bar{F}'_\rho(1) = \sigma_w^2 \int Dz \left[\phi'(\sqrt{\bar{q}^*}z)\right]^2, \qquad (20)$$

$$g_\rho \equiv \bar{F}''_\rho(1) = \sigma_w^2 \bar{q}^* \int Dz \left[\phi''(\sqrt{\bar{q}^*}z)\right]^2. \qquad (21)$$

In the smooth-class analysis below we assume $g_\rho > 0$. If $g_\rho = 0$, as for a linear activation, the quadratic Landau term is absent and the leading nonzero term determines a degenerate case rather than the generic smooth universality class. For kinked, ReLU-like activations, $\bar{F}''_\rho(1)$ is not finite and the analytic expansion below is not the correct organizing principle; we treat that universality class separately.

The Landau equation of state follows by expanding $\bar{F}_\rho(c)$ around $c = 1$. Writing $c^* = 1 - m$ with $0 < m \ll 1$ acting as the order parameter,

$$\bar{F}_\rho(1 - m) = \bar{F}_\rho(1) - \chi_\rho m + \frac{g_\rho}{2} m^2 + \mathcal{O}(m^3) \qquad (22)$$

$$= (1 - h) - \chi_\rho m + \frac{g_\rho}{2} m^2 + \mathcal{O}(m^3), \qquad (23)$$

imposing the fixed-point condition $1 - m = \bar{F}_\rho(1 - m)$ yields

$$h + tm - \frac{g_\rho}{2} m^2 = 0, \quad \Longleftrightarrow \quad h = \frac{g_\rho}{2} m^2 - tm, \qquad (24)$$

with $t \equiv \chi_\rho - 1$. The physical (nonnegative) solution is

$$m(t, h) = \frac{t + \sqrt{t^2 + 2g_\rho h}}{g_\rho}. \qquad (25)$$

At the correlation edge $t = 0$, dropout displaces the fixed point by

$$m(0, h) = \sqrt{\frac{2h}{g_\rho}}. \qquad (26)$$

For the $\alpha$ column we define the singular Landau potential $f_{\text{sing}}$ through the equation of state. In the smooth case,

$$\frac{\partial f_{\text{sing}}}{\partial m} = -h - tm + \frac{g_\rho}{2} m^2, \qquad (27)$$

and in the kinked case,

$$\frac{\partial f_{\text{sing}}}{\partial m} = -h - tm + \kappa m^{3/2}. \qquad (28)$$

---

[5]The $\rho$-dependence is implicit through $\bar{q}^*$. The full computation is given in App. A.3.

*Table 1.* Definitions of critical exponents in the thermodynamic limit ($t \to 0$, $h \to 0$).

| Exponent | Mathematical Definition |
|---|---|
| $\beta$ | $m \sim \lvert t \rvert^\beta \quad (h = 0)$ |
| $\delta$ | $m \sim \lvert h \rvert^{1/\delta} \quad (t = 0)$ |
| $\gamma$ | $\chi_M \equiv \left.\frac{\partial m}{\partial h}\right\vert_{h \to 0} \sim \lvert t \rvert^{-\gamma}$ |
| $\alpha$ | $C \equiv -\frac{\partial^2 f}{\partial t^2} \sim \lvert t \rvert^{-\alpha}$ |
| $\nu_t$ | $\xi \sim \lvert t \rvert^{-\nu_t} \quad (h = 0)$ |
| $\nu_\rho$ | $\xi \sim \lvert h \rvert^{-\nu_\rho} \quad (t = 0)$ |
| $\theta_{\text{rel}}$ | $m \sim \ell^{-\theta_{\text{rel}}} \quad (t = 0, h = 0)$ |

Evaluating $f_{\text{sing}}$ on the stable branch at $h = 0$ gives $f_{\text{sing}} \sim \lvert t \rvert^3$ for the smooth class and $f_{\text{sing}} \sim \lvert t \rvert^5$ for the kinked class, hence $\alpha = -1$ and $\alpha = -3$, respectively.

This square-root power law is our first encounter with a larger family of critical exponents familiar in the physics literature, defined in Table 1. The key exponents for our analysis are those governing the depth correlation length $\xi_{c,\rho}$, as it sets the characteristic depth scale over which distinct inputs remain distinguishable and therefore provides an effective upper bound on the depth over which the network can reliably propagate information (and learn). Their derivations are straightforward but cumbersome, so we collect them in App. B and simply quote the resulting exponent values in Table 2. For reference, we also list the standard mean-field (Landau) exponents of the Ising universality class (Ising, 1925; Landau & Lifshitz, 1980) and the corresponding mean-field exponents of spin-glass theory (Edwards & Anderson, 1975; Parisi, 1979; 1980; 1983) for comparison.

*Table 2.* Critical exponents for neural network initialization across different universality classes. The kinked $\nu_\rho$ is a normal-form field exponent; ordinary bias-free ReLU dropout follows the constrained path $t = -h$ but has the same leading $\xi \sim h^{-1/3}$ law (App. B).

| Universality Class | $\nu_t$ | $\beta$ | $\theta_{\text{rel}}$ | $\gamma$ | $\delta$ | $\nu_\rho$ | $\alpha$ |
|---|---|---|---|---|---|---|---|
| **Smooth activation** | 1 | 1 | 1 | 1 | 2 | $\frac{1}{2}$ | $-1$ |
| **Kinked activation** | 1 | 2 | 2 | 1 | $\frac{3}{2}$ | $\frac{1}{3}$ | $-3$ |
| **Spin glass (SK MF)** | $\frac{1}{2}$ | 1 | 2 | 1 | 2 | $\frac{1}{4}$ | $-1$ |
| **Ising-like** | $\frac{1}{2}$ | $\frac{1}{2}$ | 1 | 1 | 3 | $\frac{1}{3}$ | 0 |

Figure 1 summarizes these power laws from direct mean-

field recursion and shows where higher-order corrections become visible away from the asymptotic regime.

## 3.2. Scaling Collapse and Crossover Region

At leading normal-form order, dropout and detuning enter the same local equation of state. The collapse below is exact for the truncated Landau normal form, while the full mean-field recursion receives higher-order corrections such as $\mathcal{O}(m^3)$ in the smooth case, $\mathcal{O}(m^2)$ in the kinked case, and corrections from the nonlinear microscopic map $h = \Delta(\rho)$. These corrections become visible away from the asymptotic near-critical regime. From the field-theory point of view, the resulting collapse is the standard signature of criticality: in the massless limit, connected $n$-point functions become homogeneous scaling functions of their separations, so the data collapse after rescaling by the appropriate critical dimensions (Zinn-Justin, 2002). In practice, experiments are rarely tuned exactly to the correlation edge $t = 0$ while simultaneously taking the dropout field to zero, $h \to 0$. The useful near-critical description is therefore a two-parameter scaling function.

For smooth activations, the order parameter $m \equiv 1 - c^*$ obeys (24),

$$h = \frac{g_\rho}{2} m^2 - tm, \qquad t \equiv \chi_\rho - 1. \tag{29}$$

The explicit crossover solution is the physical branch in (25). Define rescaled variables

$$\tilde{m} \equiv m \sqrt{\frac{g_\rho}{2h}}, \qquad \tilde{t} \equiv -\frac{t}{\sqrt{2g_\rho h}}. \tag{30}$$

Then all curves collapse onto the universal scaling function

$$\tilde{m} = \sqrt{1 + \tilde{t}^2} - \tilde{t}. \tag{31}$$

The crossover scale is

$$|t| \sim \sqrt{g_\rho h}. \tag{32}$$

For $|t| \ll \sqrt{g_\rho h}$, one is in the field-dominated regime

$$m \sim \sqrt{\frac{2h}{g_\rho}}. \tag{33}$$

For $|t| \gg \sqrt{g_\rho h}$, the asymptotics depend on the sign of $t$:

$$t < 0: \qquad m \simeq \frac{h}{|t|}, \tag{34}$$

$$t > 0: \qquad m \simeq \frac{2t}{g_\rho} + \frac{h}{t} \cdots . \tag{35}$$

Thus, at either side of the crossover window the system is either in the linear-response regime on the subcritical side

$(t < 0)$, or it sits on the chaotic fixed-point branch on the supercritical side $(t > 0)$.

For kinked activations, the leading non-linearity is non-analytic and the equation of state takes the form (see (100)),

$$h = \kappa m^{3/2} - tm + \mathcal{O}(m^2). \tag{36}$$

This implies the scaling form

$$m(t, h) = \left( \frac{h}{\kappa} \right)^{2/3} \mathcal{F}\left( \frac{t}{\kappa^{2/3} h^{1/3}} \right). \tag{37}$$

Let

$$u \equiv \frac{t}{\kappa^{2/3} h^{1/3}}, \tag{38}$$

and define $y(u)$ as the positive real root of

$$y^3 - uy^2 - 1 = 0 \qquad \Longrightarrow \qquad \mathcal{F}(u) = y(u)^2. \tag{39}$$

The associated crossover scale is

$$|t| \sim \kappa^{2/3} h^{1/3}, \tag{40}$$

showing that kinked activations lie in a distinct universality class from smooth activations. For literal bias-free $\mathrm{ReLU}$ with standard inverted dropout, the microscopic path locks $t = \rho - 1 = -h$, so the $t = 0, h > 0$ direction is a normal-form field direction rather than a directly independent $\mathrm{ReLU}$ knob. This path nevertheless satisfies $t/(\kappa^{2/3} h^{1/3}) \to 0$, hence it enters the same field-dominated regime and retains the leading constrained-path law $\xi \sim h^{-1/3}$; App. B gives the short derivation and the nonzero-bias caveat.

Figure 2 shows this collapse for smooth activations. This also clarifies the tanh/$\mathrm{ReLU}$ dichotomy observed in ResNet mean-field theory (Yang & Schoenholz, 2017): tanh belongs to the smooth class, where the analytic $m^2$ term makes weak fields especially important and long correlation lengths are preserved when dropout is kept small; $\mathrm{ReLU}$ belongs to the kinked class, where the $m^{3/2}$ branch point broadens the crossover window and makes the dynamics more forgiving to imperfect tuning. In short, smooth channels reward keeping the dropout field very weak, while kinked channels tolerate detuning over a wider range; neither class is uniformly preferable.

## 3.3. Optimal Dropout Scheduling and Regularization Reach

So far we have taken the keep probability to be constant with depth. Since dropout enters as a relevant field that cuts off the depth correlation length even at the correlation edge, it is natural to ask how to allocate a fixed dropout budget across layers so as to preserve near-critical propagation as much as possible. Let us introduce a depth-dependent keep probability $\rho_\ell$ and define the associated dropout field

$$h_\ell \equiv \Delta(\rho_\ell), \tag{41}$$

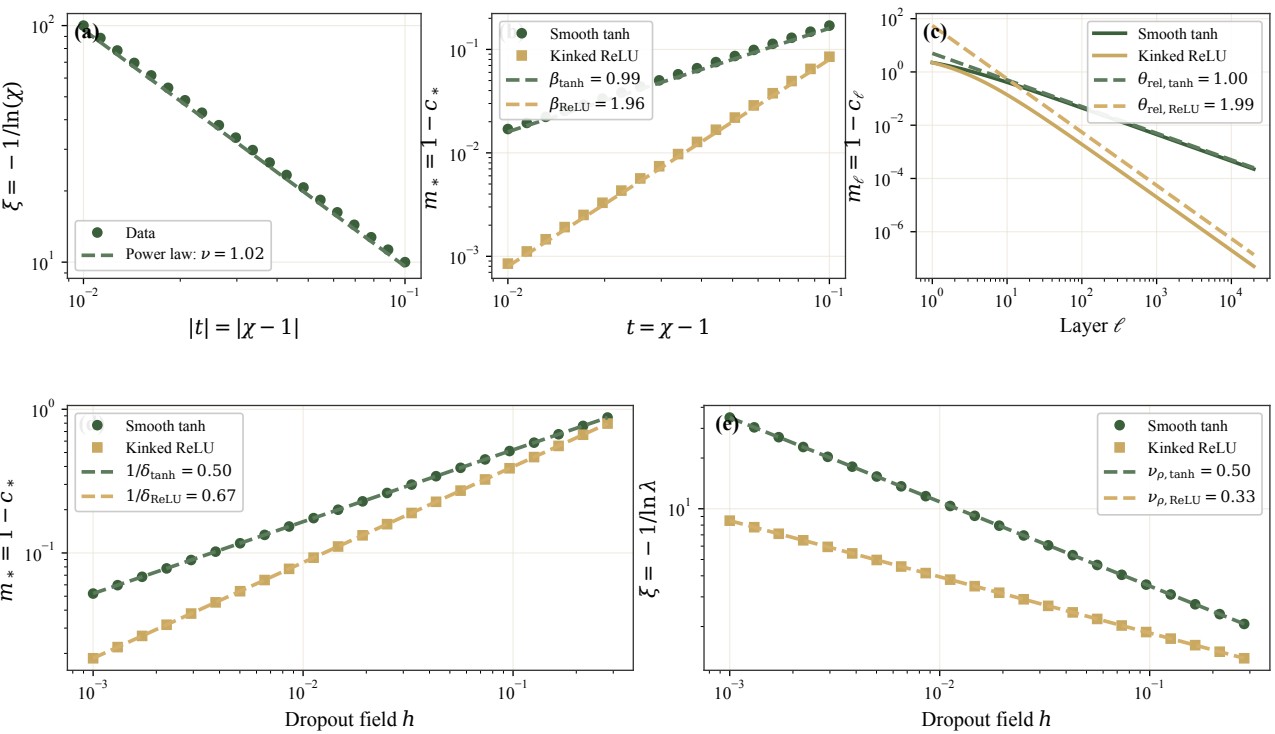

Figure 1. Critical scaling for smooth (tanh) and kinked (ReLU) activation functions, comparing tuning at zero dropout to tuning at the edge-of-chaos using a dropout field. The top row compares the different critical exponents at zero dropout and probes critical detuning decay, while the bottom row explores on critical networks with non-zero dropout. As the variables grow, higher-order effects become comparable and the linearized theory departs from the full recursion.

i.e. the shift of the correlation map at perfect alignment induced by keep probability $\rho_\ell$. In the smooth universality class, linearizing the correlation map about the dropout-shifted fixed point gives (see (85))

$$\lambda(t, h_\ell) \simeq 1 - \sqrt{t^2 + 2g_\rho h_\ell}. \tag{42}$$

For $0 < \lambda_\ell \lesssim 1$, perturbations propagate multiplicatively,

$$\delta c_L \sim \prod_{\ell=1}^{L} \lambda_\ell \delta c_0, \qquad \lambda_\ell \equiv \lambda(t, h_\ell), \tag{43}$$

which motivates defining an effective inverse correlation length as the mean decay rate per layer,

$$\xi_{\text{eff}}^{-1} \equiv -\frac{1}{L}\sum_{\ell=1}^{L}\log\lambda_\ell \simeq \frac{1}{L}\sum_{\ell=1}^{L}\sqrt{t^2 + 2g_\rho h_\ell}. \tag{44}$$

At the correlation edge $t = 0$, this reduces to

$$\xi_{\text{eff}}^{-1}(t=0) \simeq \sqrt{2g_\rho}\frac{1}{L}\sum_{\ell=1}^{L} h_\ell^{1/2} \propto \frac{1}{L}\sum_{\ell=1}^{L} h_\ell^{1/2}. \tag{45}$$

We now fix a total dropout budget and local bounds,

$$\bar{h} \equiv \frac{1}{L}\sum_{\ell=1}^{L} h_\ell, \qquad 0 \le h_\ell \le h_{\max}. \tag{46}$$

The budget above is a budget in the dropout field $h_\ell$, not directly in the raw dropout probability $p_{\text{drop},\ell} = 1 - \rho_\ell$. For weak dropout,

$$h_\ell = a p_{\text{drop},\ell} + \mathcal{O}(p_{\text{drop},\ell}^2),$$
$$a = \frac{\sigma_w^2}{q^*}\int Dz\,\phi^2(\sqrt{q^*}z), \tag{47}$$

so a fixed $h$-budget and a fixed dropout-probability budget agree to leading order. For larger dropout, the nonlinear map $h_\ell = \Delta(\rho_\ell)$ should be used explicitly. Maximizing $\xi_{\text{eff}}$ is equivalent to minimizing $\xi_{\text{eff}}^{-1}$, so in the continuum limit $(x = \ell/L)$ we obtain the constrained variational problem

$$\min_h \int_0^1 h(x)^{1/2}dx \qquad \text{s.t.} \qquad \int_0^1 h(x)dx = \bar{h}. \tag{48}$$

Since $h^{1/2}$ is concave, Jensen's inequality implies that any constant schedule maximizes $\int h^{1/2}$ at fixed mean; hence any minimizer must saturate the box constraint and is step-like. Equivalently, any schedule that takes values in $\{0, h_{\max}\}$ with active fraction $f = \bar{h}/h_{\max}$ is optimal for the mean-field functional. A front-loaded representative, selected below by regularization reach, is

$$h(x) = \begin{cases} h_{\max}, & 0 \le x \le f, \\ 0, & f < x \le 1, \end{cases} \qquad f = \frac{\bar{h}}{h_{\max}}, \tag{49}$$

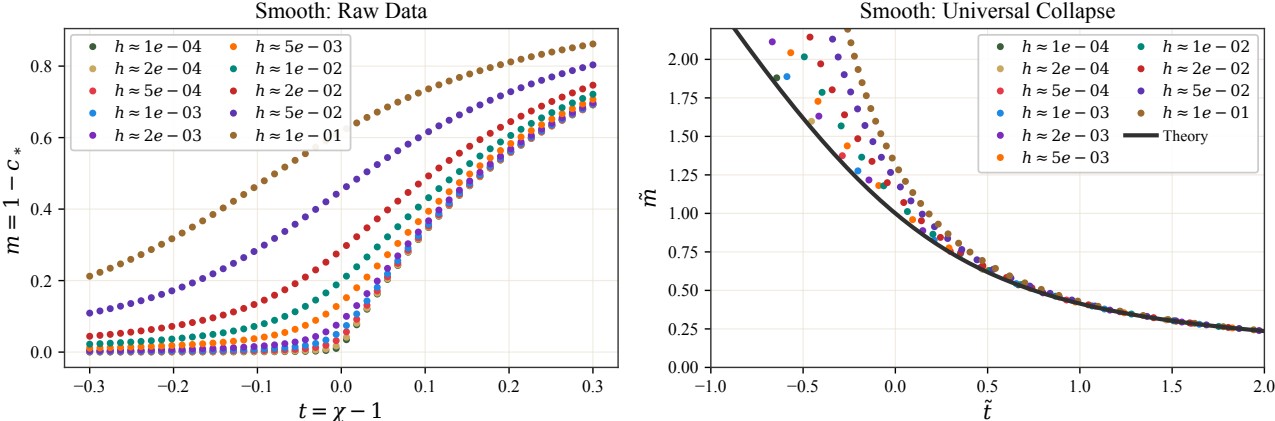

*Figure 2.* Two-parameter crossover and scaling collapse of the dropout-deformed equation of state for the smooth universality class (tanh). Plots obtained using MFT recursion relations. The curves collapse onto a universal function after rescaling by $\tilde{t}$ and $\tilde{m}$. The kinked counterpart is shown in Fig. 5.

where $x = 0$ denotes the input side. All permutations of this saturated set give the same $\xi_{\mathrm{eff}}$ in the mean-field objective; the ordering is fixed only after the regularization-reach principle is introduced. For this representative, uniform dropout yields $\xi_{\mathrm{eff}} \sim \bar{h}^{-1/2}$ while the step schedule gives

$$\frac{\xi_{\mathrm{eff,step}}}{\xi_{\mathrm{eff,uniform}}} = \sqrt{\frac{h_{\max}}{\bar{h}}} \geq 1. \qquad (50)$$

For kinked activations at $t = 0$, one has $\xi^{-1} \propto h^{1/3}$, so the objective becomes $\int_0^1 h(x)^{1/3} dx$, which is also concave and therefore the same conclusion immediately follows.

Since $\xi_{\mathrm{eff}}$ in (45) is symmetric in $\{h_\ell\}$, any permutation of a saturated schedule pays the same correlation-length cost. The tie-breaker is to spend that cost where it regularizes the most downstream layers (App. A.5). Apply dropout only at layer $\ell$. For masked and clean activations $\tilde{y}_k^{(\ell)}$ and $y_k$ at a later layer $k$, define their angular decorrelation as

$$\eta_k^{(\ell)} \equiv 1 - \frac{\mathbb{E}_p[\langle \tilde{y}_k^{(\ell)}, y_k \rangle]}{\sqrt{\mathbb{E}_p[\|\tilde{y}_k^{(\ell)}\|^2]\|y_k\|^2}}. \qquad (51)$$

Thus $\eta_k^{(\ell)} = 0$ means perfect clean–masked alignment. Immediately after the mask, $\eta_\ell^{(\ell)} = 1 - \sqrt{\rho_\ell} = (1 - \rho_\ell)/2 + \mathcal{O}((1 - \rho_\ell)^2)$, while $h_\ell = a(1 - \rho_\ell) + \mathcal{O}((1 - \rho_\ell)^2)$; hence $\eta_\ell^{(\ell)} \propto h_\ell$ to leading order. Linearizing the same wide-limit Gaussian correlation channel downstream gives $\eta_{\ell+r}^{(\ell)} \simeq C h_\ell e^{-r/\xi_c}$.

The readout perturbation $C h_\ell e^{-(L-\ell)/\xi_c}$ is larger for later dropout, so readout variance is not the front-loading objective. What matters for scheduling is downstream exposure: how much later computation trains while seeing the mask. A mask inserted early is seen, with decaying strength, by

many later blocks; a late mask has less time to decay but fewer blocks left to regularize. We therefore define $\mathcal{D}_\ell$ as the cumulative downstream exposure from placing dropout at layer $\ell$, motivating

$$\mathcal{D}_\ell \propto h_\ell \sum_{r=1}^{L-\ell} e^{-r/\xi_c}$$

$$\approx h_\ell \xi_c \left(1 - e^{-(L-\ell)/\xi_c}\right) \equiv h_\ell w_\ell. \qquad (52)$$

Since $w_\ell = \xi_c(1 - e^{-(L-\ell)/\xi_c})$ decreases with $\ell$, maximizing $\sum_{\ell=1}^{L} h_\ell w_\ell$ subject to $\sum_{\ell=1}^{L} h_\ell = L\bar{h}$ and $0 \leq h_\ell \leq h_{\max}$ is a linear program solved by filling the earliest layers first.

To test these predictions, we deliberately work in controlled regimes where mean-field assumptions remain valid (width $\gg$ depth, moderate dropout) rather than pursuing benchmark performance. We evaluate MLPs and Vision Transformers (Dosovitskiy et al., 2021) using CIFAR-10/100 (Krizhevsky, 2009) (full details in App. D), where finite-width corrections and extensive data augmentation do not obscure the signal-propagation effects central to our analysis. The clearest experimental signal is loss: Table 3 shows final cross-entropy reductions of 18–35% in the MLP and activation-sweep settings, with smaller but consistent reductions in the transformer runs. Accuracy improves as well (e.g., ViT on CIFAR-100: linearly decreasing achieves 49.38% vs 48.69% constant, $p < 0.05$), but we treat it mainly as a confirmatory metric. The appendices check that this is not a single-run effect: the gains persist across $\bar{h}$-sweeps and large-width sweeps for kinked ReLU MLPs (App. D.4, Fig. 8), across the smooth GELU activation class (App. D.4, Fig. 9), and across transformer schedules and component ablations (Figs. 10 and 12). The advantage weakens only at high dropout or in overly narrow networks,

*Table 3.* Final-epoch improvements over the corresponding constant-dropout baseline. Loss reductions are relative reductions in cross-entropy. Accuracy gains are shown as percentage-point differences and relative percent improvements over the baseline final accuracy. Detailed appendix tables list best and final test accuracy where applicable.

| Experiment | Schedule | Loss reduction | $\Delta$ acc. (pp) | Acc. gain |
|---|---|---|---|---|
| MLP overfitting, Fig. 6 | Step (early) | +17.9% | +0.83 | +2.0% |
| MLP budget controls, Fig. 7 | Big step (1/3) | +22.6% | +1.08 | +2.6% |
| ReLU $h$-sweep at $\bar{h} = 0.1$, Fig. 8 | Big step (1/3) | +35.4% | +2.04 | +5.0% |
| GELU $h$-sweep at $\bar{h} = 0.1$, Fig. 9 | Big step (1/3) | +29.8% | +0.62 | +1.5% |
| ViT CIFAR-100, Fig. 10 | Linear (decreasing) | +4.2% | +0.66 | +1.4% |
| ViT CIFAR-10 ablations, Fig. 12 | Both blocks, step (early) | +6.3% | +0.52 | +0.7% |

precisely where the theory exits its regime of validity.

## 4. Conclusions and Further Scope for Research

By extending mean-field signal propagation to include multiplicative dropout, we established that activation non-analyticity fundamentally dictates universality. This formalism yields a Landau equation of state for the decorrelation order parameter, a universal two-parameter scaling collapse, and distinct critical exponents for smooth versus kinked channels. Mean-field theory is often best at clarifying the mechanisms inside wide networks, and less often at producing directly testable design rules; here it gives one such rule. Treating dropout as a depth-dependent field leads to front-loaded schedules: the mean-field objective fixes the saturated allocation, and regularization reach places that allocation near the input. These schedules reduce test loss by 18–35% in MLP settings and by 4–6% in Vision Transformer settings relative to constant dropout at the same budget (Table 3), with consistent secondary accuracy gains and no extra computational cost. Extending this mean-field analysis to attention mechanisms and incorporating finite-width corrections are natural directions for future work.

## 5. Limitations

Our analysis is the leading-order, large-width description: finite-width corrections, controlled perturbatively by $L/N$, are not computed explicitly. They are not expected to change the smooth/kinked classification, but can accumulate with depth and induce subleading shifts to the optimal profile.

Architectural scope is also restricted. We treat dropout most explicitly for MLPs; App. A.4 argues that infinite-channel CNNs and residual branches inherit the same smooth/kinked split through the local Gaussian kernel, but we do not derive the full spatial-mode CNN Landau theory, run CNN experiments, or develop a dropout-deformed ResNet theory. Skip connections in particular modify large-depth propagation while preserving the local channel, so promoting residual-branch strength to a control parameter would put the Transformer experiments on firmer footing.

Finally, the theory is an initialization theory of *forward* correlation dynamics. We give the leading backward covariance recursion and note that dropout creates the same diagonal/off-diagonal asymmetry for gradients, but do not develop the full backward critical theory (finite-width gradient susceptibilities, training-time mask correlations, feature-learning effects), nor do we model how observables evolve after feature learning substantially reshapes representations, e.g. in catapult regimes (Zhu et al., 2024). Extending the same approach to training-time regularizers, dropout warm-up, adaptive schedules, $L^2$ penalties, or broader scaling-law questions (Bahri et al., 2024a) is a natural direction for future work.

*Acknowledgments.* The author thanks Riccardo Penco for support and discussions, and the anonymous ICML reviewers for feedback that sharpened the scheduling discussion and the treatment of the kinked-class scaling path. This work is partially supported by the U.S. Department of Energy under grant DE-SC0010118.

## Reproducibility Statement

All code, configurations, seeds, dropout schedules, and figure scripts are available in the commit-pinned repository linked in the introduction. App. D reports the run counts, hyperparameters, dataset splits, hardware, and sweep details; uncertainties are standard errors across runs. The mean-field fits use deterministic recursion code from the same repository.

## Impact Statement

The main practical benefit is improved performance at fixed training cost, or potentially reduced training cost when targeting a fixed performance level. We do not anticipate additional ethical risks.

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

# A. Mean-Field Background and Dropout Recursions

## A.1. Notation Guide

The same few letters carry several nearby meanings, so we collect the main conventions here.

*Table 4.* Notation used across the main text and appendices.

| Symbol | Meaning |
|---|---|
| $\rho_\ell$ | keep probability in layer $\ell$ |
| $p_{\mathrm{drop},\ell} = 1 - \rho_\ell$ | raw dropout probability |
| $p_j^{(\ell)}$ | Bernoulli dropout mask variable |
| $m = 1 - c^*$ | decorrelation order parameter at the fixed point |
| $m_\ell = 1 - c_\ell$ | depth-dependent decorrelation used in relaxation laws |
| $\chi_\perp, \chi_1$ | no-dropout angular susceptibility at $c = 1$ |
| $\chi_\rho$ | dropout-deformed angular susceptibility $\bar{F}'_\rho(1)$ |
| $\chi_M$ | response susceptibility $\partial m / \partial h$ |
| $\chi, \chi_q$ | generic Hermite/angle susceptibility and variance-channel susceptibility |
| $\theta_{\mathrm{rel}}$ | relaxation exponent, $m_\ell \sim \ell^{-\theta_{\mathrm{rel}}}$ |

## A.2. Mean-Field Theory Primer

Even the Standard Model of physics, with only on the order of a few dozen empirical parameters, invites the feeling that some deeper organizing principle is yet to be found. Fermi made the same point more sharply to Dyson by quoting von Neumann: "With four parameters I can fit an elephant, and with five I can make him wiggle his trunk" (Dyson, 2004). Modern machine learning pushes this worry to an extreme: parameters are abundant, so understanding cannot mean tracking each weight individually. A pressing question in modern ML literature is what set of parameters or effective fields can give a useful description of the network's behavior, and how to derive such a description from the microscopic model.

Mean-field theory gives one such effective description: instead of tracking every neuron and every random weight, one tracks a small number of self-consistent fields. In the signal-propagation version used here, those fields are variances, covariances, and correlations. This is the deep information propagation line of mean-field theory (Poole et al., 2016; Schoenholz et al., 2017), with broader background in statistical-mechanics treatments of deep learning (Bahri et al., 2020; 2024b). The same infinite-width covariance recursion can also be read as the NNGP kernel recursion (Lee et al., 2018); we use the MFT language because our questions concern fixed points, susceptibilities, correlation lengths, and critical exponents rather than the induced function-space prior or training dynamics.

Section 2 gives the main-body version of the signal-propagation story. Here we only fix notation and record the elementary steps used later in the appendices. For a depth-$L$ MLP with activation $\phi : \mathbb{R} \to \mathbb{R}$, weights and biases are initialized as

$$W_{ij}^l \sim \mathcal{N}\left(0, \frac{\sigma_w^2}{N_l}\right), \qquad b_i^l \sim \mathcal{N}\left(0, \sigma_b^2\right), \tag{53}$$

and propagated by

$$z_{i;a}^l = W_{ij}^l y_{j;a}^l + b_i^l, \qquad y_{i;a}^{l+1} = \phi\left(z_{i;a}^l\right), \tag{54}$$

where $a$ labels the input. We write

$$\int Dz (\cdots) = \frac{1}{\sqrt{2\pi}} \int_{-\infty}^{\infty} dz\, e^{-z^2/2} (\cdots), \tag{55}$$

and use the same convention for multiple Gaussian variables.

The Gaussian closure comes from the fact that each preactivation is a sum of many independently initialized weighted inputs. In the infinite-width limit this central-limit effect becomes exact layer-by-layer, while finite-width corrections are organized perturbatively in the depth-to-width aspect ratio, schematically $L/N$ (Roberts et al., 2022). The single-input variance and

two-input covariance recursions are therefore the closed equations already quoted in Sec. 2:

$$q_{aa}^l = \sigma_w^2 \int Dz \phi^2 \left( \sqrt{q_{aa}^{l-1}} z \right) + \sigma_b^2,$$

$$q_{ab}^l = \sigma_w^2 \int Dz_1 Dz_2 \phi(u_1) \phi(u_2) + \sigma_b^2,$$

$$u_1 = \sqrt{q_{aa}^{l-1}} z_1, \qquad c_{ab}^{l-1} = \frac{q_{ab}^{l-1}}{\sqrt{q_{aa}^{l-1} q_{bb}^{l-1}}},$$

$$u_2 = \sqrt{q_{bb}^{l-1}} \left( c_{ab}^{l-1} z_1 + \sqrt{1 - \left(c_{ab}^{l-1}\right)^2} z_2 \right). \tag{56}$$

Once the variance relaxes to $q^*$, these equations induce a one-dimensional correlation map $c_{ab}^l = F(c_{ab}^{l-1})$. Without dropout, perfect alignment is a fixed point, $F(1) = 1$. Its linear stability is controlled by the angular susceptibility

$$\chi_1 = \left. \frac{\partial c_{ab}^l}{\partial c_{ab}^{l-1}} \right|_{c_{ab}=1} = \sigma_w^2 \int Dz \left[ \phi'(\sqrt{q^*} z) \right]^2, \tag{57}$$

called $\chi_\perp$ in the main text and in (Roberts et al., 2022). The ordered, chaotic, and critical regimes correspond to $\chi_1 < 1$, $\chi_1 > 1$, and $\chi_1 = 1$ respectively; the last case is the edge-of-chaos, where the cross-input correlation length can diverge.

The usual He initialization follows from this criterion in the special case of ReLU. Since $\phi'(z) = \theta(z)$ almost everywhere,

$$\int Dz \left( \phi'(\sqrt{q^*} z) \right)^2 = \frac{1}{2}, \qquad \chi_1 = 1 \implies \sigma_w^2 = 2, \tag{58}$$

which is the variance-$2/N$ initialization. The same example also shows why variance preservation and correlation criticality are not identical: for ReLU,

$$q_{aa}^l = \frac{\sigma_w^2}{2} q_{aa}^{l-1} + \sigma_b^2, \tag{59}$$

so with nonzero bias the finite-$q^*$ condition requires $\sigma_w^2 < 2$, while exact angular criticality sets $\sigma_w^2 = 2$. In practice one either sets biases to zero or works slightly subcritical to keep the variance scale finite.

Finally, the correlation length used throughout the paper is obtained by linearizing the correlation map around its fixed point. If $r_c = F'(c^*)$, then

$$|c^* - c_{ab}^l| \propto e^{-l/\xi_c}, \qquad \xi_c^{-1} = -\log|r_c|. \tag{60}$$

The analogous variance length $\xi_q$ tracks relaxation of $q_{aa}^l$ to $q^*$, but the edge-of-chaos phenomenon is primarily controlled by $\xi_c$: it measures how many layers preserve distinctions between different inputs. This is the scale deformed by dropout in the main analysis.

### A.3. Derivation of Correlation Map and Derivatives with Dropout

This subsection supplies the steps leading to Eqs. (20) and (21), starting from the two-input correlation recursion that underlies the perfect-alignment shift $\bar{F}_\rho(1) = 1 - \Delta$ in (15).

Fix a layer where the single-input variance has converged to the dropout-shifted fixed point $\bar{q}^*$ defined by (13). For two inputs, let $(u_1, u_2)$ be jointly Gaussian with

$$\mathbb{E}[u_1^2] = \mathbb{E}[u_2^2] = \bar{q}^*, \qquad \mathbb{E}[u_1 u_2] = c\bar{q}^*, \qquad c \in [-1, 1]. \tag{61}$$

The normalized correlation map is

$$\bar{F}_\rho(c) \equiv \frac{\bar{q}_{ab}^{l+1}}{\bar{q}^*} = \frac{\sigma_w^2 \mathbb{E}[\phi(u_1)\phi(u_2)] + \sigma_b^2}{\bar{q}^*}. \tag{62}$$

At $c = 1$, $u_1 = u_2$ and $\bar{F}_\rho(1) = 1 - \Delta$ by definition, giving (15).

Since $\bar{q}^*$ is independent of $c$, differentiating (62) yields

$$\bar{F}'_\rho(c) \;=\; \frac{\sigma_w^2}{\bar{q}^*} \frac{\partial}{\partial c} \mathbb{E}\big[\phi(u_1)\phi(u_2)\big]. \tag{63}$$

Write $u_i = \sqrt{\bar{q}^*} v_i$, where $(v_1, v_2)$ are standard Gaussians with $\mathbb{E}[v_1 v_2] = c$, and define

$$G(c) \;\equiv\; \mathbb{E}\Big[\phi\big(\sqrt{\bar{q}^*} v_1\big)\phi\big(\sqrt{\bar{q}^*} v_2\big)\Big]. \tag{64}$$

Price's theorem (Price, 1958) gives

$$\frac{\mathrm{d}}{\mathrm{d}c} G(c) \;=\; \mathbb{E}\left[ \frac{\partial^2}{\partial v_1 \partial v_2}\Big(\phi\big(\sqrt{\bar{q}^*} v_1\big)\phi\big(\sqrt{\bar{q}^*} v_2\big)\Big)\right]. \tag{65}$$

A direct differentiation yields

$$\frac{\partial^2}{\partial v_1 \partial v_2}\Big(\phi\big(\sqrt{\bar{q}^*} v_1\big)\phi\big(\sqrt{\bar{q}^*} v_2\big)\Big) = \bar{q}^* \phi'\big(\sqrt{\bar{q}^*} v_1\big)\phi'\big(\sqrt{\bar{q}^*} v_2\big). \tag{66}$$

Therefore,

$$G'(c) = \bar{q}^* \mathbb{E}\Big[\phi'\big(\sqrt{\bar{q}^*} v_1\big)\phi'\big(\sqrt{\bar{q}^*} v_2\big)\Big]. \tag{67}$$

Taking $c \to 1$ collapses $v_1 = v_2 = z$ with $z \sim \mathcal{N}(0, 1)$, hence

$$G'(1) = \bar{q}^* \int Dz \big[\phi'(\sqrt{\bar{q}^*} z)\big]^2. \tag{68}$$

Substituting into (63) gives

$$\bar{F}'_\rho(1) = \sigma_w^2 \int Dz \big[\phi'(\sqrt{\bar{q}^*} z)\big]^2, \tag{69}$$

which is Eq. (20). Differentiating (63) once more:

$$\bar{F}''_\rho(c) \;=\; \frac{\sigma_w^2}{\bar{q}^*} \frac{\partial^2}{\partial c^2} \mathbb{E}\big[\phi(u_1)\phi(u_2)\big]. \tag{70}$$

Applying Price's theorem again to $f(v_1, v_2) = \phi'(\sqrt{\bar{q}^*} v_1)\phi'(\sqrt{\bar{q}^*} v_2)$ yields

$$\frac{\mathrm{d}}{\mathrm{d}c} \mathbb{E}\Big[\phi'(\sqrt{\bar{q}^*} v_1)\phi'(\sqrt{\bar{q}^*} v_2)\Big] = \bar{q}^* \mathbb{E}\Big[\phi''(\sqrt{\bar{q}^*} v_1)\phi''(\sqrt{\bar{q}^*} v_2)\Big]. \tag{71}$$

Combining with the previous step gives

$$\frac{\partial^2}{\partial c^2} \mathbb{E}\big[\phi(u_1)\phi(u_2)\big] = (\bar{q}^*)^2 \mathbb{E}\Big[\phi''(\sqrt{\bar{q}^*} v_1)\phi''(\sqrt{\bar{q}^*} v_2)\Big]. \tag{72}$$

Evaluating at $c = 1$ again collapses $v_1 = v_2 = z$, so

$$\frac{\partial^2}{\partial c^2} \mathbb{E}\big[\phi(u_1)\phi(u_2)\big]\bigg|_{c=1} = (\bar{q}^*)^2 \int Dz \big[\phi''(\sqrt{\bar{q}^*} z)\big]^2. \tag{73}$$

Substituting into (70) yields

$$\bar{F}''_\rho(1) = \sigma_w^2 \bar{q}^* \int Dz \big[\phi''(\sqrt{\bar{q}^*} z)\big]^2, \tag{74}$$

which is Eq. (21).

## A.4. Extensions to Infinite-Channel CNNs and ResNets

We briefly discuss extensions to architectures not treated in the main analysis. Although our derivation is for MLPs, the universality-class mechanism is tied to the local Gaussian kernel rather than to the MLP architecture itself. Other architectures introduce additional structure that can change the exact exponents, but the qualitative smooth-versus-kinked distinction should remain when the same local channel controls the wide-limit recursion.

For example, the same local channel which appears in MLPs re-emerges in infinite-width CNNs when the number of channels is taken to infinity: preactivations at different spatial positions become jointly Gaussian across channels, and the covariance recursion composes the same bivariate Gaussian activation expectation with a linear operator that averages over convolutional filter offsets (LeCun et al., 1998; Xiao et al., 2018). In schematic form, a spatial covariance $K_{\alpha\beta}^\ell(x, x')$ evolves as $K^{\ell+1} = \sigma_b^2 + \sigma_w^2 \mathcal{A}[V_\phi(K^\ell)]$, where $V_\phi$ is the scalar Gaussian channel analyzed above and $\mathcal{A}$ is the convolutional averaging operator. Independent dropout shifts the single-input variance entering $V_\phi$ as in the fully connected case, while the cross-input covariance is still controlled by the same bivariate expectation. Therefore, for an isolated leading spatial mode, convolution changes the relevant eigenvalue and mode shape but not the local smooth-versus-kinked non-analyticity: smooth activations retain the analytic Landau equation, while ReLU-like kinks retain the $m^{3/2}$ branch point. A complete CNN theory would promote $m$ to a spatial covariance-mode amplitude and track boundary conditions, pooling, and degeneracies of $\mathcal{A}$.

ResNets modify depth dynamics in a complementary way: skip connections can change exponential convergence to subexponential or polynomial behavior, and norms may drift rather than relax to a fixed point (Yang & Schoenholz, 2017). Still, the residual branch evaluates the same nonlinear Gaussian channel $V_\phi$, so the local analytic distinction is unchanged. The tanh/ReLU asymptotic split observed in (Yang & Schoenholz, 2017) is consistent with our smooth/kinked classes; a dropout-deformed ResNet theory would additionally track residual-branch strength and norm drift.

## A.5. Regularization-Reach Derivation

The mean-field propagation objective is invariant under permutations of the depth profile $h_\ell$, so it fixes the amount of dropout but not its ordering. The ordering can be derived from the same correlation-length physics by asking how much downstream computation sees a dropout perturbation before it is absorbed.

First apply dropout only at layer $\ell$. Write $\tilde{y}_k^{(\ell)}$ for the masked trajectory and $y_k$ for the clean trajectory. We measure the size of the perturbation at layer $k$ by the clean-vs.-masked angular decorrelation

$$\eta_k^{(\ell)} \equiv 1 - \frac{\mathbb{E}_p[\langle \tilde{y}_k^{(\ell)}, y_k \rangle]}{\sqrt{\mathbb{E}_p[\|\tilde{y}_k^{(\ell)}\|^2]\|y_k\|^2}}. \tag{75}$$

Here $\eta_k^{(\ell)} = 0$ means the two trajectories are perfectly aligned at layer $k$. Immediately after the mask,

$$\eta_\ell^{(\ell)} = 1 - \sqrt{\rho_\ell} = \frac{1 - \rho_\ell}{2} + \mathcal{O}((1 - \rho_\ell)^2). \tag{76}$$

The dropout field used in the main text is the same small parameter up to a constant:

$$h_\ell = a(1 - \rho_\ell) + \mathcal{O}((1 - \rho_\ell)^2), \qquad a = \frac{\sigma_w^2}{q^*} \int Dz\, \phi^2(\sqrt{q^*}z), \tag{77}$$

so $\eta_\ell^{(\ell)} \propto h_\ell$ to leading order. This constant of proportionality does not affect the scheduling optimizer.

Next, we propagate this perturbation downstream. For $k > \ell$, the clean-vs.-masked pair is governed by the same wide-limit Gaussian correlation channel as a two-input pair, up to variance-relaxation corrections. Linearizing around the relevant fixed point gives

$$\eta_{\ell+r}^{(\ell)} \simeq C h_\ell e^{-r/\xi_c}, \tag{78}$$

where $r$ is the number of downstream layers and $C$ is independent of $h_\ell$. If we only looked at the final readout, we would get $C h_\ell e^{-(L-\ell)/\xi_c}$. This is larger for later dropout, because late perturbations have fewer layers over which to decay. That is why readout variance is not the object that explains front-loading.

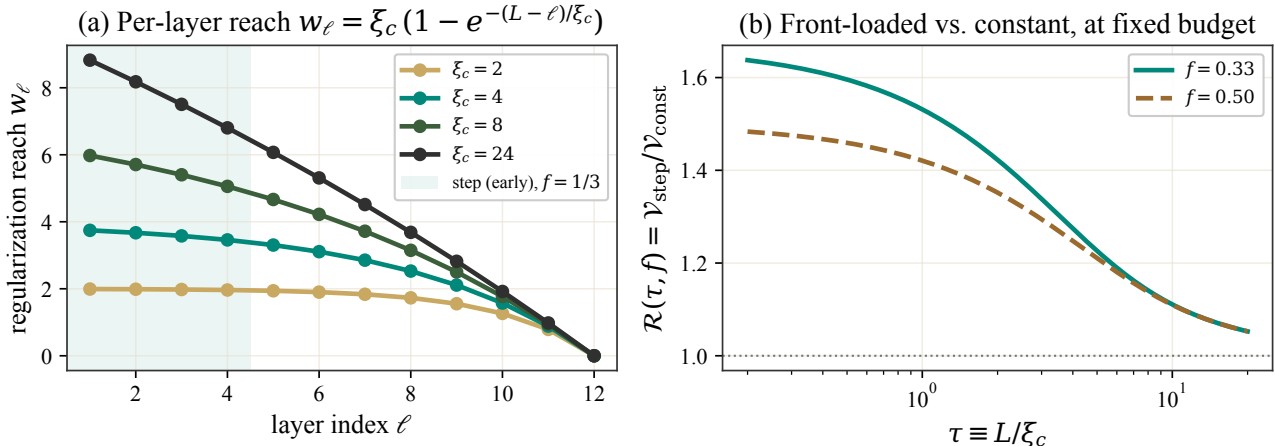

*Figure 3.* Regularization-reach weights at fixed network depth $L = 12$. (a) Per-layer weight $w_\ell = \xi_c(1 - e^{-(L-\ell)/\xi_c})$ is monotone decreasing in $\ell$ for any $\xi_c$; the linear program over the dropout budget therefore fills the smallest-$\ell$ layers first (shaded: step-early with $f = 1/3$). For $L \ll \xi_c$ the reach is essentially linear in $L - \ell$; for $L \gg \xi_c$ it saturates at $\xi_c$. (b) Ratio $\mathcal{R}(\tau, f) = \mathcal{V}_{\text{step}}/\mathcal{V}_{\text{const}}$ of front-loaded to constant schedules at fixed budget, with $\tau \equiv L/\xi_c$. The advantage peaks near $\tau \sim 1$ and decays as $\tau$ grows, qualitatively matching the larger MLP gains and smaller ViT gains in Table 3.

For scheduling, the relevant object is not the perturbation left at the readout, but the amount of downstream computation exposed to the mask. Each later layer trains on the perturbed representation in proportion to how much of the mask-induced decorrelation remains. We denote this cumulative exposure by $\mathcal{D}_\ell$. It is not additional dropout strength; it is the downstream sum of the remaining perturbations, proportional to $\sum_{r=1}^{L-\ell} \eta_{\ell+r}^{(\ell)}$.

Substituting the exponential decay into this exposure sum gives

$$\mathcal{D}_\ell \propto h_\ell \sum_{r=1}^{L-\ell} e^{-r/\xi_c} \approx h_\ell \int_0^{L-\ell} e^{-s/\xi_c}\, ds = h_\ell \xi_c \left(1 - e^{-(L-\ell)/\xi_c}\right) \equiv h_\ell w_\ell. \tag{79}$$

The approximation replaces the discrete sum by its continuum integral; using the exact geometric sum changes only constants and leaves the same monotone ordering. The weight $w_\ell = \xi_c(1 - e^{-(L-\ell)/\xi_c})$ decreases with $\ell$: earlier masks are visible to more downstream layers before they are absorbed (Fig. 3).

The mean-field objective says that every permutation of the same saturated dropout set pays the same correlation-length cost. We therefore choose the permutation that gives the largest downstream regularization reach, which is what motivates using dropout in the first place:

$$\max_{\{h_\ell\}} \sum_{\ell=1}^{L} h_\ell w_\ell \qquad \text{s.t.} \qquad \sum_{\ell=1}^{L} h_\ell = L\bar{h}, \qquad 0 \le h_\ell \le h_{\max}. \tag{80}$$

This is a linear program. Since $w_1 > w_2 > \cdots > w_L$, it is solved by filling the largest weights first:

$$h_\ell^\star = \begin{cases} h_{\max}, & 1 \le \ell \le fL, \\ 0, & fL < \ell \le L, \end{cases} \qquad f = \frac{\bar{h}}{h_{\max}}. \tag{81}$$

Thus the front-loaded step schedule follows from the same correlation length that controls signal propagation. Empirically, sliding a fixed saturated dropout block $B$ from early to late layers should degrade performance according to $\sum_{\ell \in B} w_\ell$, with the slope controlled by $\xi_c$.

## B. Critical Exponents and Non-Analyticity

The Hermite decomposition in App. C.3 diagnoses which expansion is valid here. In the Gaussian channel,

$$F(c) = \frac{\sigma_w^2 \sum_{n \ge 0} a_n^2 c^n + \sigma_b^2}{q^*}. \tag{82}$$

Rapid Hermite decay makes this series analytic at $c = 1$, giving the ordinary Taylor/Landau equation below. A kink produces a power-law Hermite tail, so high-degree modes contribute collectively as $c \to 1$; the Taylor expansion is replaced by the branch-point term $m^{3/2}$, which is the source of the kinked exponents.

## B.1. Smooth Networks

The standard static exponents follow directly from (24). Along the critical isotherm $t = 0$,

$$h = \frac{g_\rho}{2} m^2 \qquad \Rightarrow \qquad m \sim h^{1/2}, \tag{83}$$

so $\delta = 2$. The analogue of the magnetic susceptibility is $\chi_M(t) \equiv \partial m/\partial h|_{h\to0}$. On the subcritical side $t < 0$, the analytic branch satisfies $m \simeq -h/t$, hence $\chi_M(t) \sim |t|^{-1}$ and $\gamma = 1$. Along $h = 0$, the physical branch is $m = 0$ for $t \leq 0$ and $m = 2t/g_\rho$ for $t \geq 0$, so $m \sim t$ as $t \to 0^+$ and $\beta = 1$.

The depth correlation length follows from linearizing the correlation dynamics about the displaced fixed point. To leading order,

$$\lambda \equiv \bar{F}'_\rho(c^*) = \bar{F}'_\rho(1 - m) \simeq \chi_\rho - g_\rho m = 1 + t - g_\rho m, \tag{84}$$

and substituting (25) gives

$$\lambda(t, h) = 1 - \sqrt{t^2 + 2g_\rho h}. \tag{85}$$

For $\lambda \simeq 1$ with $\lambda > 0$, the depth correlation length satisfies

$$\xi_{c,\rho}^{-1} \equiv -\log \lambda \simeq 1 - \lambda = \sqrt{t^2 + 2g_\rho h}. \tag{86}$$

Thus $\xi_{c,\rho}(t, 0) \sim |t|^{-1}$ and $\nu_t = 1$, while $\xi_{c,\rho}(0, h) \sim h^{-1/2}$ and $\nu_\rho = 1/2$. Equivalently, $y_t = 1/\nu_t = 1$ and $y_\rho = 1/\nu_\rho = 2$, so dropout is the more relevant perturbation of the correlation edge.

At the marginal point $t = h = 0$, the same smooth normal form gives the depth relaxation law:

$$m_{\ell+1} - m_\ell \simeq -\frac{g_\rho}{2} m_\ell^2. \tag{87}$$

In a continuum-depth approximation this gives $m(\ell) \sim \ell^{-1}$, hence $\theta_{\rm rel} = 1$ for the smooth class.

A pseudo-free-energy functional consistent with (24) is

$$f(t, m; h) = -\frac{t}{2} m^2 + \frac{g_\rho}{6} m^3 - hm, \tag{88}$$

since $\partial f/\partial m = 0$ is equivalent to (24). This object should be read in the same spirit as a low-order one-particle-irreducible (1PI) effective action in the sense that it is built so itsthe stationarity condition gives the equation of state, while its on-shell singular part organizes the thermodynamic-style exponents; analytic $m$-independent backgrounds can be added without changing the physics of the recursion.

At $h = 0$, substituting the physical branch $m = 0$ for $t \leq 0$ and $m = 2t/g_\rho$ for $t \geq 0$ yields

$$f_{\rm on}(t, 0) = 0 \quad (t \leq 0), \qquad f_{\rm on}(t, 0) = -\frac{2}{3g_\rho^2} t^3 \quad (t \geq 0). \tag{89}$$

Therefore the analogue of the specific heat, $C \equiv -\partial_t^2 f_{\rm on}$, stays finite and vanishes linearly:

$$C(t) = \frac{4}{g_\rho^2} t\Theta(t) \sim |t|, \tag{90}$$

where $\Theta(t)$ is the Heaviside step function. In standard exponent language $C_{\rm sing}(t) \sim |t|^{-\alpha}$, this corresponds to

$$\alpha = -1, \tag{91}$$

up to an additive analytic background $f_0(t, h)$ that does not affect the equation of state.

Collecting exponents for the smooth universality class:

$$\beta = 1, \qquad \gamma = 1, \qquad \delta = 2, \qquad \nu_t = 1, \qquad \nu_\rho = \frac{1}{2}, \qquad \theta_{\text{rel}} = 1, \qquad \alpha = -1. \tag{92}$$

For comparison, standard Ising-like Landau mean-field theory (with $m \to -m$ symmetry and a quartic interaction) has

$$\beta = \frac{1}{2}, \qquad \gamma = 1, \qquad \delta = 3, \qquad \nu = \frac{1}{2}, \qquad \alpha = 0. \tag{93}$$

Our dropout-deformed correlation-edge theory shares $\gamma = 1$, but the lack of a $\mathbb{Z}_2$ symmetry (and the resulting cubic interaction) shifts $\beta$ and $\delta$, produces $\nu_t = 1$ rather than $\nu = 1/2$, and yields a specific-heat analogue that vanishes at the transition ($\alpha = -1$) rather than exhibiting the Ising mean-field behavior ($\alpha = 0$).

### B.2. Kinked Networks

Up to now, the Landau-style expansion of $\bar{F}_\rho(c)$ around $c = 1$ relied on the assumption that $\bar{F}_\rho(c)$ is analytic in

$$m \equiv 1 - c. \tag{94}$$

For smooth activations this is the natural situation, and the first nonlinear correction is of order $m^2$, controlled by

$$g_\rho \propto \int Dz \left[ \phi''(\sqrt{\bar{q}^*}z) \right]^2. \tag{95}$$

For kinked activations, this logic fails: the map remains smooth for $c < 1$, but as it approaches $c \to 1$, it is governed by a branch point, and the leading nonlinear correction is non-analytic in $m$. This changes the scaling of the dropout deformation.

We illustrate this explicitly for the ReLU activation $\phi(x) = \max(0, x)$. To keep formulas compact we set $\sigma_b = 0$ throughout this subsection and use the same inverted-dropout convention as before, with masks drawn independently across inputs.

The mean-field correlation map for ReLU admits the standard arc-cosine closed form (Cho & Saul, 2009). Writing $c \in [-1, 1]$ for the preactivation correlation at some depth,

$$F_{\text{ReLU}}(c) = \frac{1}{\pi} \left[ \sqrt{1 - c^2} + \left( \pi - \cos^{-1} c \right) c \right]. \tag{96}$$

Near perfect alignment, setting $c = 1 - m$ with $0 < m \ll 1$, one obtains the non-analytic expansion

$$F_{\text{ReLU}}(1 - m) = 1 - m + \kappa m^{3/2} + \mathcal{O}(m^2), \tag{97}$$

$$\kappa \equiv \frac{2\sqrt{2}}{3\pi}.$$

Thus, the Taylor expansion valid for smooth activations would have missed the $m^{3/2}$ contribution.

We now turn on dropout and again view dropout as a field deformation at $c = 1$:

$$\bar{F}_\rho(1) = 1 - h, \qquad h > 0. \tag{98}$$

In the present $\sigma_b = 0$ setting, this field is exactly

$$h = 1 - \bar{F}_\rho(1) = 1 - \rho, \tag{99}$$

independent of the activation, since at $c = 1$ the only source of mismatch is the independence of the two dropout masks. We write the fixed point as $c^* = 1 - m^*$ with $m^* > 0$ and impose $c^* = \bar{F}_\rho(c^*)$. For the literal bias-free ReLU map, $\bar{F}_\rho(c) = \rho F_{\text{ReLU}}(c)$, so $\chi_\rho = \rho$ and $t = \rho - 1 = -h$. Thus the $t = 0, h > 0$ direction below is the kinked normal-form field direction, while standard ReLU dropout follows a constrained path through the same scaling function.

Near $m^* \ll 1$, the kinked analogue of the Landau equation of state is obtained by combining the shift $\bar{F}_\rho(1) = 1 - h$, the linear slope $\chi_\rho = 1 + t$, and the non-analytic term (97):

$$h = \kappa m^{3/2} - tm + \mathcal{O}(m^2), \qquad t \equiv \chi_\rho - 1. \tag{100}$$

Along the normal-form critical isotherm $t = 0$, this gives

$$m^* \sim \left(\frac{h}{\kappa}\right)^{2/3}, \qquad (t = 0), \tag{101}$$

so the smooth exponent $\delta = 2$ is replaced by

$$m \propto h^{1/\delta_{\text{kink}}}, \qquad \delta_{\text{kink}} = \frac{3}{2}. \tag{102}$$

And with this we immediately find the smoking gun telling us that kinked and smooth activations fall into distinct mean-field universality classes under the dropout deformation. As before, the depth scale follows from linearizing the correlation dynamics around $c^*$. Differentiating (97) gives

$$\lambda \equiv \bar{F}'_\rho(c^*) \simeq 1 + t - \frac{3\kappa}{2}\sqrt{m^*}. \tag{103}$$

At $t = 0$ we therefore have

$$1 - \lambda \propto \sqrt{m^*} \propto h^{1/3}. \tag{104}$$

Using $\xi_{c,\rho}^{-1} \simeq 1 - \lambda$ for $\lambda \simeq 1$ yields

$$\xi_{c,\rho}(0,h) \sim h^{-1/3}, \qquad \nu_{\rho,\text{kink}} = \frac{1}{3}, \tag{105}$$

to be compared with the smooth result $\xi_{c,\rho}(0,h) \sim h^{-1/2}$. For ordinary bias-free ReLU dropout, substituting the constrained path $t = -h$ into (100) gives

$$h = \kappa m^{3/2} + hm + \mathcal{O}(m^2). \tag{106}$$

Since the leading solution has $m \sim h^{2/3}$, the $hm$ term is subleading. Moreover,

$$1 - \lambda \simeq h + \frac{3\kappa}{2}\sqrt{m} \sim h^{1/3}, \tag{107}$$

so the same $\nu_{\rho,\text{kink}} = 1/3$ survives as a constrained-path exponent.

The remaining kinked exponents follow from the same equation of state. At $h = 0$ and $t > 0$,

$$\kappa m^{3/2} = tm \qquad \Rightarrow \qquad m = \left(\frac{t}{\kappa}\right)^2, \tag{108}$$

so $\beta_{\text{kink}} = 2$. On the subcritical side $t < 0$, the small-field balance is $h \simeq |t|m$, giving $\chi_M \sim |t|^{-1}$ and therefore $\gamma_{\text{kink}} = 1$. Substituting the zero-field branch into (103) gives $1 - \lambda \propto t$, so $\xi_{c,\rho}(t,0) \sim t^{-1}$ and $\nu_{t,\text{kink}} = 1$.

Adding nonzero ReLU bias variance does not restore a finite-$q^*$ critical isotherm. With inverted dropout,

$$\bar{q}_{\ell+1} = \frac{\sigma_w^2}{2\rho}\bar{q}_\ell + \sigma_b^2, \tag{109}$$

so finite variance requires $a \equiv \sigma_w^2/(2\rho) < 1$, while angular criticality requires $\chi_\rho = \sigma_w^2/2 = 1$, incompatible with $\rho < 1$. Equivalently, $h = a(1-\rho)$ and $t = \rho a - 1 = -h - (1-a)$, so finite variance adds a subcritical detuning that cuts off the asymptotic dropout law unless one takes the singular double scaling $1 - a \ll h^{1/3}$.

It is also useful, in direct analogy with the smooth case, to package (100) into a pseudo-free-energy. A functional whose stationarity condition reproduces the kinked equation of state is

$$f_{\text{kink}}(t,m;h) = -\frac{t}{2}m^2 + \frac{2\kappa}{5}m^{5/2} - hm, \tag{110}$$

since $\partial f_{\text{kink}}/\partial m = 0$ gives $h = \kappa m^{3/2} - tm$ at leading order. In the effective-action language, the $m^{5/2}$ term is the pseudo-free-energy avatar of the branch point in the ReLU correlation map; this is precisely where the kinked universality

class departs from the analytic Landau polynomial above.[6] At zero field the physical branch is $m = 0$ for $t \leq 0$ and $m = (t/\kappa)^2$ for $t \geq 0$, so the on-shell free energy behaves as

$$f_{\text{kink,on}}(t, 0) = 0 \quad (t \leq 0), \qquad f_{\text{kink,on}}(t, 0) = -\frac{1}{10\kappa^4}t^5 \quad (t \geq 0). \tag{111}$$

Thus the analogue of the specific heat, $C \equiv -\partial_t^2 f_{\text{kink,on}}$, stays finite and in fact vanishes as

$$C(t) = \frac{2}{\kappa^4}t^3\Theta(t) \sim |t|^3. \tag{112}$$

In the standard exponent convention $C_{\text{sing}}(t) \sim |t|^{-\alpha}$, this corresponds to

$$\alpha_{\text{kink}} = -3. \tag{113}$$

Although we used ReLU to compute the near-alignment expansion in closed form, the resulting fractional power is not a peculiarity of ReLU. Rather, it is a generic feature of the mean-field Gaussian channel whenever the activation has an ordinary kink: $\phi$ is continuous and piecewise $C^1$ with a finite jump in slope at some point. In that case the correlated-Gaussian expectation defining the normalized correlation map remains smooth for $c < 1$, but its approach to $c \to 1$ is controlled by a square-root branch associated with the thin Gaussian tube around the diagonal $u_1 = u_2$. Writing $c = 1 - m$ with $0 < m \ll 1$, the transverse fluctuations scale as $\sqrt{m}$, so the probability that two nearly aligned preactivations fall on opposite sides of the kink scales as $\mathcal{O}(\sqrt{m})$; in that straddling region a finite-slope kink produces activation mismatches of size $\mathcal{O}(\sqrt{m})$, so the leading non-linear correction to the kernel scales as $\mathcal{O}(\sqrt{m}) \times \mathcal{O}(m) = \mathcal{O}(m^{3/2})$. Thus, for any standard kinked activation one expects

$$F(1 - m) = 1 - m + \kappa_\phi m^{3/2} + \mathcal{O}(m^2), \tag{114}$$

with an activation-dependent coefficient $\kappa_\phi$, while the exponent $3/2$ is fixed by the tube geometry.

Finally, it is useful to recall what happens when dropout is switched off exactly at the marginal point. Setting

$$\rho = 1, \qquad t = 0, \tag{115}$$

the non-analytic correction controls the approach to $c = 1$. Writing $m_\ell \equiv 1 - c_\ell$ and using (97),

$$m_{\ell+1} - m_\ell \simeq -\kappa m_\ell^{3/2}. \tag{116}$$

In a continuum-depth approximation $m_{\ell+1} - m_\ell \to \partial_\ell m$, we obtain

$$\partial_\ell m \simeq -\kappa m^{3/2} \qquad \Rightarrow \qquad m(\ell) \sim \ell^{-2}, \tag{117}$$

which reproduces the characteristic $1 - c_\ell \sim \ell^{-2}$ relaxation of ReLU activations at the edge-of-chaos (Hayou et al., 2019) and gives $\theta_{\text{rel}} = 2$.

### B.3. Non-Analytic Expansion of the ReLU Correlation Map Near $c = 1$

Here we derive the non-analytic expansion

$$F_{\text{ReLU}}(1 - m) = 1 - m + \kappa m^{3/2} + \mathcal{O}(m^2), \qquad \kappa = \frac{2\sqrt{2}}{3\pi}, \tag{118}$$

starting from the closed form ReLU correlation map (Cho & Saul, 2009)

$$F_{\text{ReLU}}(c) = \frac{1}{\pi}\left[\sqrt{1 - c^2} + (\pi - \cos^{-1} c)c\right], \qquad c \in [-1, 1]. \tag{119}$$

Set $c = 1 - m$ with $0 < m \ll 1$.

---

[6]As before, $f_{\text{kink}}$ is defined only up to addition of an $m$-independent analytic background $f_0(t, h)$, which does not affect the equation of state.

First expand the square root term,

$$\sqrt{1 - c^2} = \sqrt{1 - (1-m)^2} = \sqrt{2m - m^2} = \sqrt{2m}\sqrt{1 - \frac{m}{2}}$$

$$= \sqrt{2m}\left(1 - \frac{m}{4} + \mathcal{O}(m^2)\right) = \sqrt{2}m^{1/2} - \frac{\sqrt{2}}{4}m^{3/2} + \mathcal{O}(m^{5/2}). \tag{120}$$

Next expand $\cos^{-1}(1-m)$. Write $\theta = \cos^{-1}(1-m)$ and use

$$\cos\theta = 1 - \frac{\theta^2}{2} + \frac{\theta^4}{24} + \mathcal{O}(\theta^6), \tag{121}$$

together with an ansatz $\theta = am^{1/2} + bm^{3/2} + \mathcal{O}(m^{5/2})$. Matching $\cos\theta = 1 - m$ order by order gives $a = \sqrt{2}$ and $b = \sqrt{2}/12$, hence

$$\cos^{-1}(1-m) = \sqrt{2}m^{1/2} + \frac{\sqrt{2}}{12}m^{3/2} + \mathcal{O}(m^{5/2}). \tag{122}$$

Now expand the product term in (119),

$$\left(\pi - \cos^{-1}c\right)c = \left(\pi - \cos^{-1}(1-m)\right)(1-m)$$

$$= \left(\pi - \sqrt{2}m^{1/2} - \frac{\sqrt{2}}{12}m^{3/2} + \mathcal{O}(m^{5/2})\right)(1-m)$$

$$= \pi(1-m) - \sqrt{2}m^{1/2} + \frac{11\sqrt{2}}{12}m^{3/2} + \mathcal{O}(m^2). \tag{123}$$

Adding (120) and (123) cancels the $\mathcal{O}(m^{1/2})$ terms and yields

$$\sqrt{1-c^2} + \left(\pi - \cos^{-1}c\right)c = \pi(1-m) + \left(-\frac{\sqrt{2}}{4} + \frac{11\sqrt{2}}{12}\right)m^{3/2} + \mathcal{O}(m^2)$$

$$= \pi(1-m) + \frac{2\sqrt{2}}{3}m^{3/2} + \mathcal{O}(m^2). \tag{124}$$

Dividing by $\pi$ gives (118) with $\kappa = 2\sqrt{2}/(3\pi)$.

The key structural point is the cancellation of the $\mathcal{O}(m^{1/2})$ pieces, leaving a leading non-analytic correction of order $m^{3/2}$, which encodes the kink.

## B.4. Origin of the $m^{3/2}$ Term for Kinked Activations

The $m^{3/2}$ term is a near-alignment effect. Write $c = 1 - m$ with $m \ll 1$. Then the two Gaussian preactivations differ only by a small transverse fluctuation. Away from a kink, the activation is locally smooth and the usual Taylor expansion produces only integer powers of $m$. The non-analytic term comes from the small set of samples where the two nearly identical preactivations fall on opposite sides of the kink. For ReLU, the main text gives the closed-form expansion

$$F(1-m) = 1 - m + \kappa m^{3/2} + O(m^2), \tag{125}$$

with $\kappa = \frac{2\sqrt{2}}{3\pi}$ for ReLU. The coefficient is specific to ReLU and to the normalization convention, but the exponent is not: the same $3/2$ power appears for any activation with an ordinary finite-slope kink.

Let $(u_1, u_2)$ be jointly Gaussian with

$$\mathbb{E}[u_1] = \mathbb{E}[u_2] = 0, \qquad \mathbb{E}[u_1^2] = \mathbb{E}[u_2^2] = 1, \qquad \mathbb{E}[u_1 u_2] = c. \tag{126}$$

The normalized correlation map takes the form

$$F(c) \propto \mathbb{E}\left[\phi(u_1)\phi(u_2)\right], \tag{127}$$

with proportionality fixed by the variance normalization used in the main text.

We study the approach to perfect alignment by writing

$$c = 1 - m, \qquad 0 < m \ll 1. \tag{128}$$

Introduce the sum and difference coordinates

$$s = \frac{u_1 + u_2}{\sqrt{2}}, \qquad d = \frac{u_1 - u_2}{\sqrt{2}}. \tag{129}$$

A direct covariance computation gives

$$\mathrm{Var}(s) = 1 + c = 2 - m, \qquad \mathrm{Var}(d) = 1 - c = m, \qquad \mathbb{E}[sd] = 0. \tag{130}$$

Hence as $m \to 0$ one has the scaling

$$s \sim O(1), \qquad d \sim O(\sqrt{m}). \tag{131}$$

Thus the average coordinate $s$ remains order one, while the separation $d$ has standard deviation $\sqrt{m}$. Geometrically, the joint Gaussian measure concentrates in a thin tube of transverse thickness $\sqrt{m}$ around the diagonal $u_1 = u_2$. Only this tube can see the kink: the probability of straddling the kink is of order $\sqrt{m}$, and inside that region the activation mismatch produced by the slope jump is also of order $\sqrt{m}$. In the kernel this gives an order-$m$ local correction carried by an order-$\sqrt{m}$ region, hence the leading non-analytic contribution scales as $m^{3/2}$.

This argument is not special to ReLU. Near the kink, any activation with finite one-sided slopes can be separated into a smooth part plus a multiple of ReLU. If those slopes are $a_\pm$ at $u = 0$, then locally

$$\phi(u) = a_- u + (a_+ - a_-)\, \mathrm{ReLU}(u) + r(u), \qquad r(u) = o(u) \text{ as } u \to 0, \tag{132}$$

where $r$ is continuous and has no slope jump at $u = 0$. This remainder behaves like an ordinary smooth contribution in the near-alignment expansion and therefore gives only integer powers of $m$. The first fractional-power term is inherited from the ReLU component, so every ordinary kink gives the same $m^{3/2}$ exponent. Only the amplitude depends on nonuniversal details such as the slope jump $a_+ - a_-$ and the precise variance normalization.

## C. Hermite and Variational Perspective

We start from the variational problem that first led us to the smooth/kinked split: choose the shape of the activation so as to maximize the mean-field correlation length. Since wide-network preactivations are Gaussian, this is naturally a problem in a function space with Gaussian measure. Hermite polynomials are the standard orthonormal basis for such spaces, so expanding the rescaled activation in Hermite modes turns the correlation-length calculation into a spectral problem. This appendix makes that reduction explicit.

Before doing this, we remove a trivial scale freedom: $\phi \mapsto a\phi$ can be absorbed by $\sigma_w^2 \mapsto \sigma_w^2/a^2$ without changing the mean-field fixed point. To focus on shape rather than scale, we fix a variance fixed point $q^*$ and the bias variance $\sigma_b^2$, and determine $\sigma_w^2$ from the fixed-point condition. With this convention, the remaining freedom is the shape of a function $f \in L^2(Dz)$ with Gaussian inner product

$$Dz \equiv \frac{e^{-z^2/2}}{\sqrt{2\pi}} dz, \qquad \langle f, g \rangle \equiv \int Dz f(z) g(z). \tag{133}$$

### C.1. A Scale-Fixed Variational Problem for $\chi$

We define the fixed-point rescaled activation

$$f(z) \equiv \phi(\sqrt{q^*} z). \tag{134}$$

Assuming $\phi$ is weakly differentiable with $f \in H^1(Dz)$, Price's theorem (Price, 1958) yields the correlation susceptibility used in deep information propagation (Schoenholz et al., 2017)

$$\chi \equiv F'(1) = \frac{\sigma_w^2}{q^*} \int Dz \big(f'(z)\big)^2. \tag{135}$$

The variance fixed-point equation

$$q^* = \sigma_w^2 \int Dz f(z)^2 + \sigma_b^2 \tag{136}$$

determines

$$\sigma_w^2 = \frac{q^* - \sigma_b^2}{\int Dz f^2}. \tag{137}$$

Substituting gives the scale-fixed factorization

$$\chi = \left(1 - \frac{\sigma_b^2}{q^*}\right) \frac{\int Dz (f')^2}{\int Dz f^2} \equiv \left(1 - \frac{\sigma_b^2}{q^*}\right) \mathcal{Q}[f], \qquad \mathcal{Q}[f] \equiv \frac{\|f'\|_2^2}{\|f\|_2^2}. \tag{138}$$

Thus, at fixed $(q^*, \sigma_b^2)$, extremizing $\chi$ over activations reduces to extremizing the Rayleigh quotient (see for example (Horn & Johnson, 1985)) $\mathcal{Q}[f]$ over the shape of $f$. Criticality $\chi = 1$ is equivalent to the constraint

$$\mathcal{Q}[f] = \left(1 - \frac{\sigma_b^2}{q^*}\right)^{-1}. \tag{139}$$

In this convention, $\mathcal{Q}[f]$ quantifies the activation's effective sensitivity to typical Gaussian fluctuations: shifting spectral weight to higher modes increases $\|f'\|_2$ relative to $\|f\|_2$ and therefore increases $\chi$ unless $\sigma_w^2$ is retuned.

### C.2. The Ornstein–Uhlenbeck Eigenvalue Problem

Because $\mathcal{Q}[f]$ is scale-invariant, we may impose the normalization $\int Dz f^2 = 1$ and extremize $\int Dz (f')^2$. Introducing a Lagrange multiplier $\lambda$, consider

$$\mathcal{S}[f] \equiv \int Dz \Big[ (f'(z))^2 - \lambda f(z)^2 \Big]. \tag{140}$$

Gaussian-weighted integration by parts (with boundary terms vanishing under mild decay assumptions) gives the Euler–Lagrange equation

$$\left(-\partial_z^2 + z \partial_z\right) f(z) = \lambda f(z), \tag{141}$$

i.e. the Ornstein–Uhlenbeck eigenproblem on $L^2(Dz)$. Its spectrum is

$$\lambda_n = n, \qquad n = 0, 1, 2, \ldots, \tag{142}$$

and the eigenfunctions form a complete orthonormal basis. Under the unit-norm constraint, these eigenfunctions are stationary points of $\mathcal{Q}[f]$, with $\mathcal{Q}[f] = \lambda$.

### C.3. Hermite Basis and Diagonal Correlation Propagation

A natural orthonormal eigenbasis of (141) is given by the normalized Hermite polynomials; this is the usual Hermite/Ornstein–Uhlenbeck diagonalization of Gaussian channels. Hermite expansions have also been used directly to express neural-network covariance recursions in the infinite-width setting (Arratia et al., 2020).

$$h_n(z) \equiv \frac{1}{\sqrt{2^n n!}} H_n\left(\frac{z}{\sqrt{2}}\right), \qquad \langle h_n, h_m \rangle = \delta_{nm}, \tag{143}$$

where $H_n$ are the physicists' Hermite polynomials.[7] They satisfy

$$h_n'(z) = \sqrt{n} h_{n-1}(z), \qquad \left(-\partial_z^2 + z \partial_z\right) h_n(z) = n h_n(z), \qquad \mathcal{Q}[h_n] = n. \tag{144}$$

Moreover, this basis diagonalizes correlated-Gaussian expectations: if $(z_1, z_2)$ are jointly Gaussian with unit variances and correlation $c \in [-1, 1]$, then

$$\mathbb{E}\big[h_n(z_1) h_m(z_2)\big] = c^n \delta_{nm}. \tag{145}$$

Thus each Hermite degree propagates independently through the Gaussian channel with attenuation $c^n$.

---

[7]Equivalently, $h_n(z) = \mathrm{He}_n(z)/\sqrt{n!}$ in terms of the probabilists' Hermite polynomials $\mathrm{He}_n$.

Expanding an arbitrary $f \in L^2(Dz)$ in this basis,

$$f(z) = \sum_{n \geq 0} a_n h_n(z), \qquad a_n = \langle f, h_n \rangle, \qquad \sum_{n \geq 0} a_n^2 = \int Dz f^2, \tag{146}$$

we obtain

$$\mathbb{E}\big[f(z_1)f(z_2)\big] = \sum_{n \geq 0} a_n^2 c^n. \tag{147}$$

Consequently, the correlation map takes the form

$$F(c) = \frac{\sigma_w^2 \sum_{n \geq 0} a_n^2 c^n + \sigma_b^2}{q^*}, \qquad q^* = \sigma_w^2 \sum_{n \geq 0} a_n^2 + \sigma_b^2. \tag{148}$$

Analyticity of $F(c)$ near $c = 1$ is controlled by the decay of $|a_n|$: rapid decay (typical for smooth activations) yields an ordinary Taylor expansion around $c = 1$, while slow decay (typical for kinked or piecewise smooth activations) allows infinitely many high-degree modes to contribute collectively and can produce non-Taylor behavior (e.g. fractional powers or logarithms) as $c \to 1$, corresponding to a branch point at $c = 1$.

Using $h_n' = \sqrt{n} h_{n-1}$, we can compute the susceptibility

$$\chi = F'(1) = \frac{\sigma_w^2}{q^*} \sum_{n \geq 1} n a_n^2. \tag{149}$$

In the zero-bias case $\sigma_b^2 = 0$ this becomes

$$\chi = \frac{\sum_{n \geq 1} n a_n^2}{\sum_{n \geq 0} a_n^2} = \mathcal{Q}[f], \tag{150}$$

so $\chi$ is the mean Hermite degree under weights proportional to $a_n^2$, and criticality $\chi = 1$ corresponds to unit mean degree.

For completeness, the variance-channel susceptibility $\chi_q \equiv \partial q^{(\ell+1)} / \partial q^{(\ell)} |_{q^*}$ can be written in the same basis as

$$\chi_q = \frac{\sigma_w^2}{q^*} \left( \sum_{n \geq 1} n a_n^2 + \sum_{n \geq 0} \sqrt{(n+1)(n+2)} a_n a_{n+2} \right). \tag{151}$$

Thus $\chi$ depends only on the distribution $\{a_n^2\}$ across Hermite degrees, whereas $\chi_q$ is additionally sensitive to relative signs/phases within each parity subsector through the $a_n a_{n+2}$ couplings.

Pure modes $f \propto h_n$ are stationary points with $\mathcal{Q}[h_n] = n$. Excluding the degenerate constant mode, or imposing $\langle f \rangle = 0$, the $n = 1$ mode is the lowest eigenmode, but by itself it gives a linear network. Nonlinear activations necessarily mix in higher Hermite degrees. Those modes raise $\mathcal{Q}[f]$, so at fixed $q^*$ and $\sigma_b^2$ they shorten the correlation depth unless $\sigma_w^2$ is reduced accordingly.

### C.4. Smooth and Kinked Activations

For smooth activations, the Hermite coefficients typically decay rapidly and $F(c)$ is analytic near $c = 1$, with $\chi$ dominated by the lowest few degrees. For kinked activations such as ReLU, $\phi'(u) = \theta(u)$ almost everywhere, so

$$\chi = \sigma_w^2 \int Dz \, \theta(u) = \frac{\sigma_w^2}{2}, \qquad \chi = 1 \Rightarrow \sigma_w^2 = 2. \tag{152}$$

*Table 5.* Orthonormal Hermite coefficients $a_n = \int Dz f(z) h_n(z)$ for two standard activations, shown for the unscaled choices $f(z) = \mathrm{ReLU}(z)$ and $f(z) = \tanh(z)$. For $f(z) = \tanh(sz)$, use $a_n(s) = \int Dz \tanh(sz) h_n(z)$, with $a_{2k}(s) = 0$ by odd parity.

| $n$ | $h_n(z)$ parity | $a_n$ for $\mathrm{ReLU}(z)$ | $a_n$ for $\tanh(z)$ |
|---|---|---|---|
| 0 | even | $\frac{1}{\sqrt{2\pi}}$ | 0 |
| 1 | odd | $\frac{1}{2}$ | 0.60570551 |
| 2 | even | $\frac{1}{2\sqrt{\pi}}$ | 0 |
| 3 | odd | 0 | $-0.14843719$ |
| 4 | even | $-\frac{1}{\sqrt{48\pi}}$ | 0 |
| 5 | odd | 0 | 0.06254752 |
| 6 | even | $\frac{1}{4\sqrt{10\pi}}$ | 0 |
| 7 | odd | 0 | $-0.03144542$ |
| 8 | even | $-\frac{15}{\sqrt{80640\pi}}$ | 0 |
| 9 | odd | 0 | 0.01741993 |
| 10 | even | $\frac{105}{\sqrt{7257600\pi}}$ | 0 |
| 11 | odd | 0 | $-0.01029184$ |

## C.5. Hermite Decompositions for ReLU and tanh

Throughout, $Z \sim \mathcal{N}(0,1)$ and

$$Dz \equiv \frac{dz}{\sqrt{2\pi}} e^{-z^2/2}. \tag{153}$$

We expand the fixed point rescaled activation $f(z)$ in an orthonormal Hermite basis of $L^2(Dz)$,

$$f(z) = \sum_{n \geq 0} a_n h_n(z), \qquad a_n = \int Dz f(z) h_n(z), \qquad \int Dz h_n(z) h_m(z) = \delta_{nm}. \tag{154}$$

With physicists Hermites $H_n$,

$$H_n(x) \equiv (-1)^n e^{x^2} \frac{d^n}{dx^n} e^{-x^2}, \qquad h_n(z) \equiv \frac{1}{\sqrt{2^n n!}} H_n\left(\frac{z}{\sqrt{2}}\right). \tag{155}$$

### C.5.1. ReLU

Let $f(z) = \mathrm{ReLU}(z) = \max(0, z)$. The coefficients have a closed form.

$$a_0 = \frac{1}{\sqrt{2\pi}}, \qquad a_1 = \frac{1}{2}, \qquad a_{2k+1} = 0 \text{ for } k \geq 1, \tag{156}$$

and for $k \geq 1$,

$$a_{2k} = \frac{(-1)^{k-1}(2k-3)!!}{\sqrt{2\pi (2k)!}}. \tag{157}$$

We use the convention $(-1)!! = 1$. If instead $f(z) = \mathrm{ReLU}(\sqrt{q^*}z)$, then all coefficients scale as $a_n \mapsto \sqrt{q^*} a_n$.

### C.5.2. tanh

Let $f(z) = \tanh(sz)$ with $s > 0$. The coefficients are

$$a_n(s) = \int Dz \tanh(sz) h_n(z). \tag{158}$$

Parity gives $a_{2k}(s) = 0$. There is no simple closed form for general $s$, but the coefficients are easy to compute numerically by Gaussian quadrature using Mathematica, or other specialized packages.

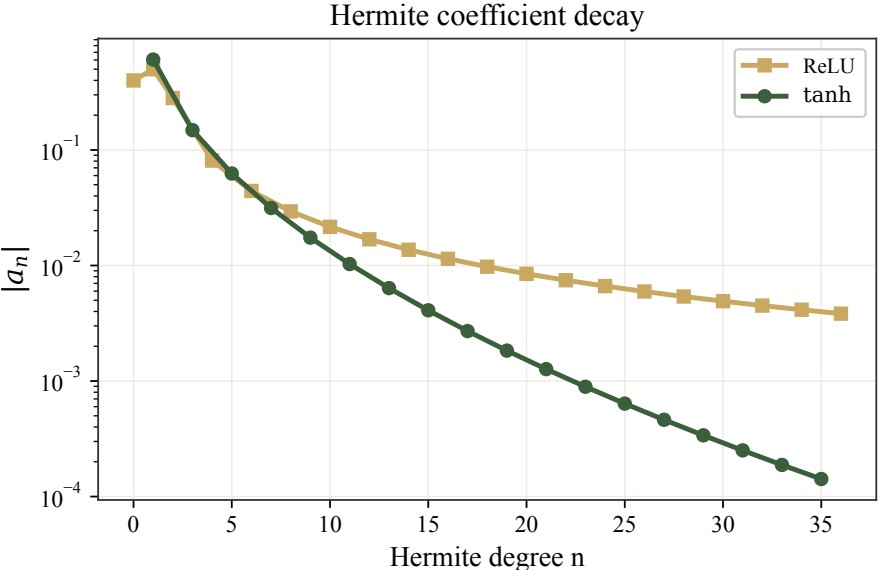

*Figure 4.* Magnitude of Hermite coefficients $|a_n|$ for ReLU versus tanh. ReLU exhibits a slow (power-law) decay, reflecting multi-scale support across Hermite degrees, while tanh decays rapidly and concentrates most spectral mass in the lowest modes.

## D. Experimental Details and Additional Figures

This appendix collects the experimental details and additional figures supporting the scheduling predictions. Code for the experiments and figure generation is available in the commit-pinned `dropout-universality-experiments` repository.

### D.1. Mean-Field Fits and Scaling Collapse

Table 6 reports exponents extracted from iterated MFT recursions.

*Table 6.* Critical exponents from log–log linear fits (dashed lines) in Fig. 1 compared with mean-field theory predictions.

| Exponent | Iterated recursion | Linear approximation |
|---|---|---|
| $\nu$ | $1.0190 \pm 0.0014$ | 1 |
| $\beta_{\text{tanh}}$ | $0.9879 \pm 0.0007$ | 1 |
| $\beta_{\text{ReLU}}$ | $1.9642 \pm 0.0019$ | 2 |
| $\theta_{\text{rel,tanh}}$ | $1.0048 \pm 0.0000$ | 1 |
| $\theta_{\text{rel,ReLU}}$ | $1.9870 \pm 0.0001$ | 2 |
| $1/\delta_{\text{tanh}}$ | $0.5171 \pm 0.0007$ | 1/2 |
| $1/\delta_{\text{ReLU}}$ | $0.6656 \pm 0.0001$ | 2/3 |
| $\nu_{\rho,\text{tanh}}$ | $0.4716 \pm 0.0014$ | 1/2 |
| $\nu_{\rho,\text{ReLU}}$ | $0.3528 \pm 0.0010$ | 1/3 |

The same scaling-collapse diagnostic can be carried out for the kinked universality class, using ReLU as the representative activation.

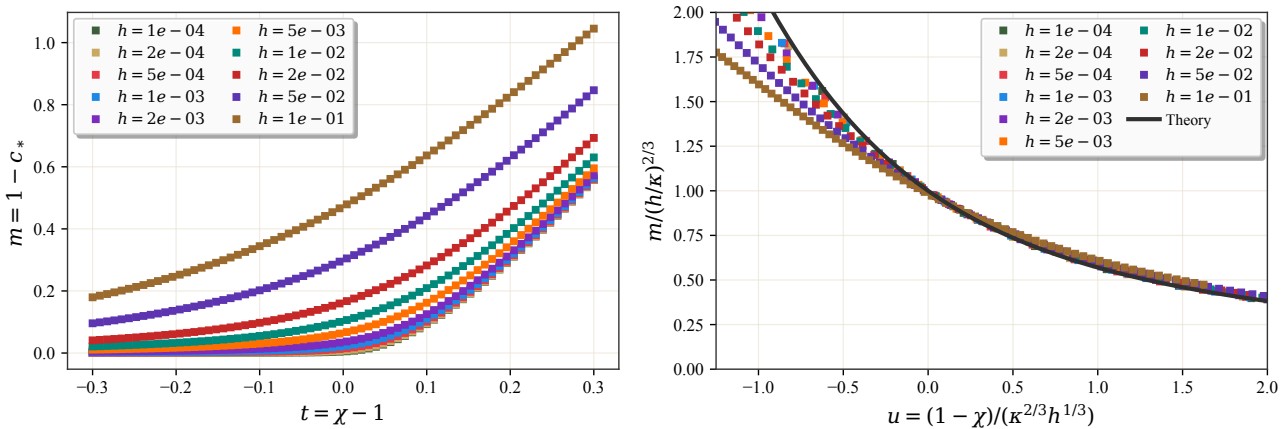

*Figure 5.* Kinked counterpart to Fig. 1, showing the corresponding universal scaling collapse.

## D.2. Dropout Scheduling Experiments

The scheduling experiments below test Sec. 3.3: MLP schedules in Fig. 6 and Tab. 7, matched-budget controls in Fig. 7 and Tab. 8, robustness sweeps in Fig. 8, ViT CIFAR-100 schedules in Fig. 10 and Tab. 11, and transformer ablations in Fig. 12 and Tab. 13.

## D.3. MLPs

Finite-width experiments support these spatial scheduling predictions. The main finding is that concentrating dropout in early layers, where a mask has the largest downstream regularization reach, yields the best generalization. The theoretically motivated schedules improve over constant dropout at no extra computational cost.

To test this prediction in a controlled finite-width setting, we design an experiment in the overfitting regime, where information propagation and regularization are both important factors in determining performance. In the effective-theory treatment of (Roberts et al., 2022), deviations from Gaussian mean-field behavior (Neal, 1996; Lee et al., 2018) generate corrections to the correlation recursions that are suppressed by $1/N$ (with $N$ the layer width) and accumulate across depth as $L/N$. At the same time, deep information propagation suggests that successful learning is controlled by a modest multiple of the correlation length (Schoenholz et al., 2017).

We train on 5000 CIFAR-10 images with width 256 and depth $L = 6$, roughly $2\xi$ for the constant-dropout baseline, using a batch size of 75 to ensure the regularizing effect of dropout through injected stochastic noise. We initialize close to the edge-of-chaos at $\sigma_w^2 = 1.98$ and $\sigma_b^2 = 0.02$, near standard He initialization but avoiding the instabilities due to the divergence of the variance fixed point $q^*$ as $\sigma_w^2 \to 2$.

We compare six depth profiles at fixed mean dropout budget: (i) no dropout, (ii) uniform dropout, (iii) a linearly increasing schedule, (iv) a linearly decreasing schedule, (v) a step schedule concentrated in the final half (Step (late)), and (vi) a step schedule concentrated in the first half (Step (early)), as summarized in Fig. 6. Front-loaded dropout consistently outperforms back-loaded dropout: early masks are visible to a longer downstream suffix before they decay over the correlation length, while late masks have little computation left to regularize. The effective correlation length heuristic remains predictive; constant dropout is most damaging, and regularization reach breaks the permutation symmetry of Eq. (45) in favor of early regularization.

The budget-control experiment checks a simpler explanation: early-concentrated dropout may only be winning because it raises the local dropout rate, not because the allocation over depth matters. After all, the step schedule applies dropout at rate $2\bar{h}$ to the first half of the network, so perhaps the same improvement could be achieved by applying this higher rate uniformly throughout.

To disentangle allocation from total dropout strength, we compare constant schedules with fields $\bar{h}, 2\bar{h}, 3\bar{h}$ against step schedules using $2\bar{h}$ in the first half or $3\bar{h}$ in the first third of the network (Fig. 7, Tab. 8).

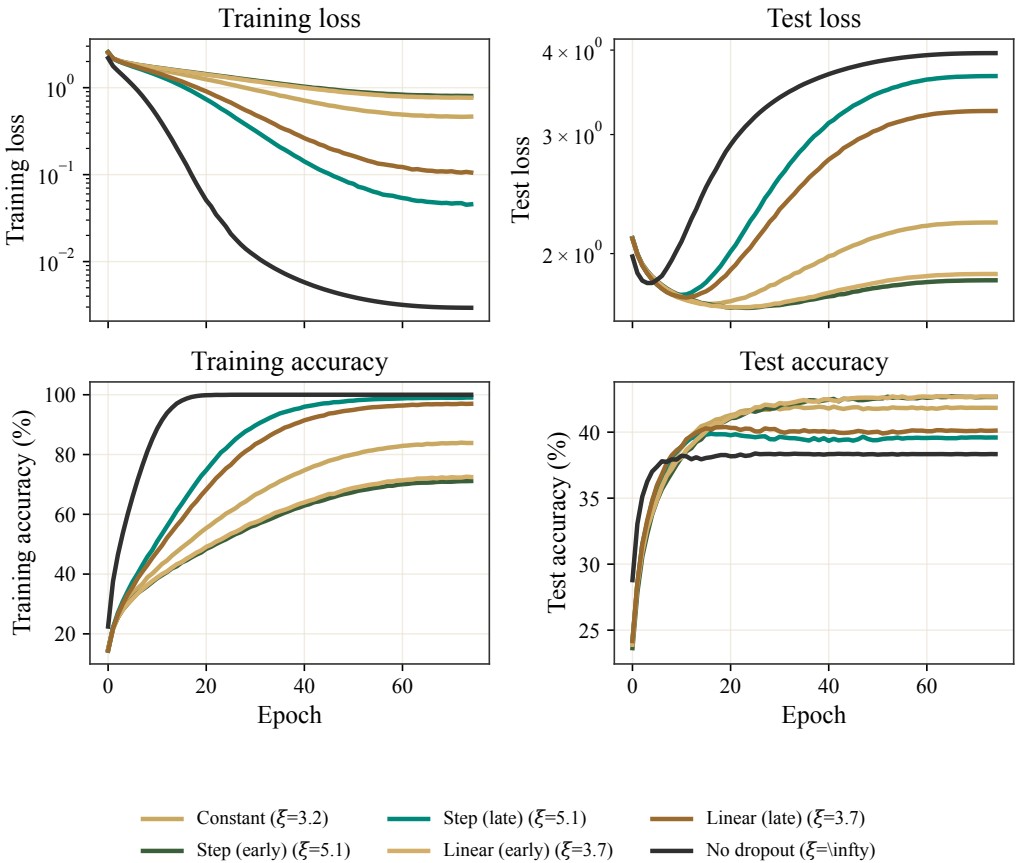

*Figure 6.* Finite-width MLP training and test curves at fixed mean dropout budget $\bar{h}$ (with $0 \le h_\ell \le h_{\max}$). We compare uniform dropout, linear ramps (increasing/decreasing with depth), and step schedules that concentrate dropout in either the first or last half of the network, together with the no-dropout baseline.

*Table 7.* Performance of different dropout schedules in the overfitting regime for MLPs and CIFAR-10, corresponding to Fig. 6.

| Schedule | $\xi_{\mathrm{eff}}$ | $L/\xi_{\mathrm{eff}}$ | Best Test Acc. (%) | Final Test Acc. (%) |
|---|---|---|---|---|
| Constant | 3.20 | 1.87 | $42.54 \pm 0.10$ | $41.84 \pm 0.10$ |
| Step (late) | 5.09 | 1.18 | $40.56 \pm 0.11$ | $39.60 \pm 0.13$ |
| Step (early) | 5.09 | 1.18 | $43.13 \pm 0.07$ | $42.67 \pm 0.09$ |
| Linear (increasing) | 3.73 | 1.61 | $41.01 \pm 0.07$ | $40.12 \pm 0.12$ |
| Linear (decreasing) | 3.73 | 1.61 | $43.11 \pm 0.09$ | $42.69 \pm 0.10$ |
| No dropout | $\infty$ | 0 | $38.77 \pm 0.10$ | $38.34 \pm 0.12$ |

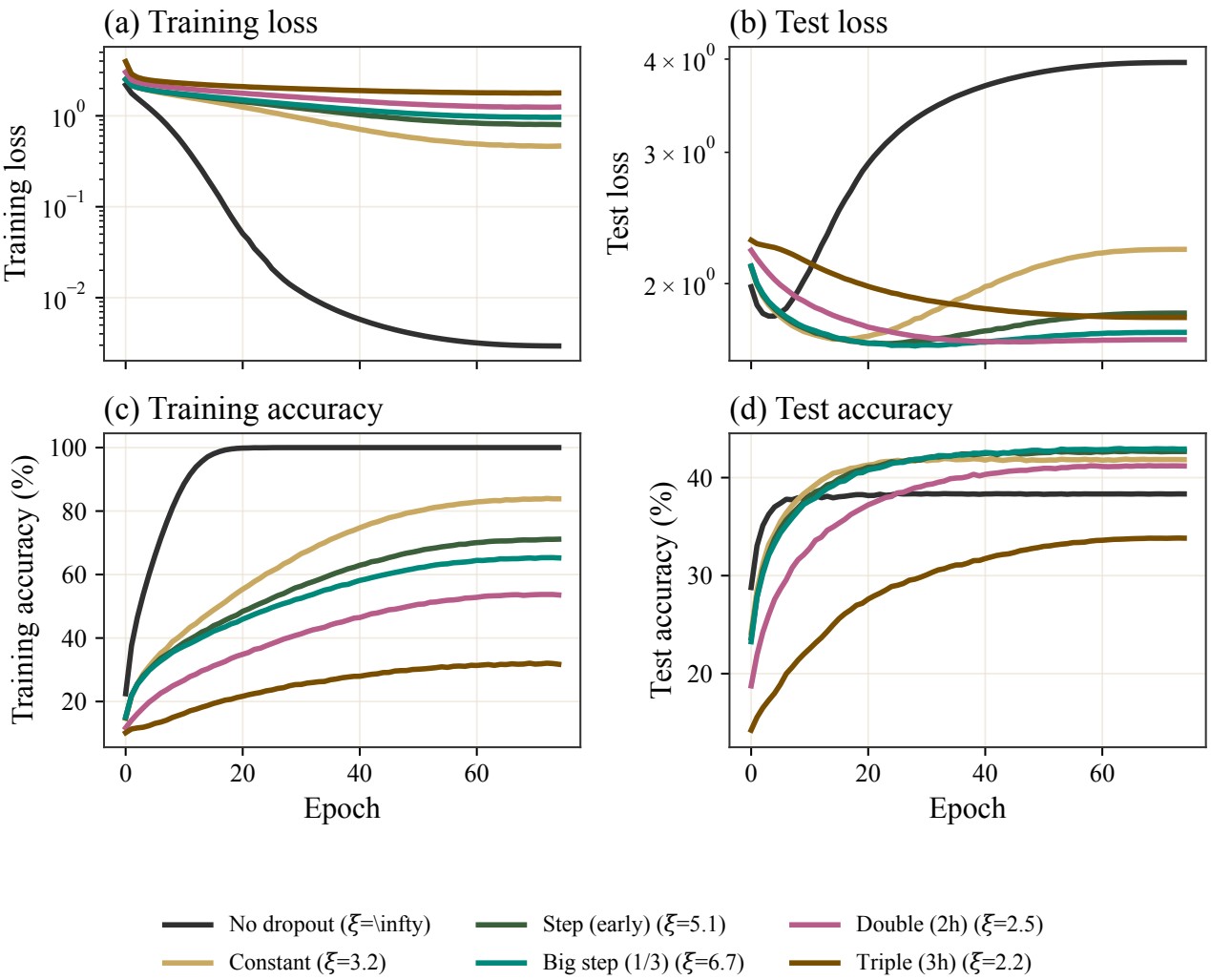

*Figure 7.* Matched-budget controls comparing front-loaded step schedules against constant dropout fields at $\bar{h}$, $2\bar{h}$, and $3\bar{h}$.

If the step schedules succeed merely because they apply locally higher dropout rates, then the uniform schedules with matching dropout should perform at least as well. In contrast, if spatial allocation genuinely matters, the step schedules should outperform their uniform counterparts despite having lower total dropout. The results in Fig. 7 and Table 8 confirm that spatial allocation dominates: Big step ($\xi_{\text{eff}} = 6.7$) achieves the highest accuracy despite having only one-third the total dropout of the schedule with $3\bar{h}$, and front-loaded step ($\xi_{\text{eff}} = 5.1$) outperforms Double despite half the total dropout budget. Moreover, increasing uniform dropout beyond $2\bar{h}$ becomes counterproductive; Triple ($3\bar{h}$) performs worst among all dropout schedules, as excessive regularization throughout the network impairs learning capacity. These findings validate the effective correlation length $\xi_{\text{eff}}$ as a predictive metric: schedules with longer $\xi_{\text{eff}}$ consistently achieve better generalization, regardless of their total dropout budget.

We note that our theoretical analysis is perturbative in $h$, with higher-order corrections of $O(h^2)$ becoming significant as dropout increases. This motivates our choice to test up to $3\bar{h}$ with $\bar{h} = 0.1$; beyond this point, the perturbative framework becomes less reliable and additional nonlinear effects may dominate. We also tested these effects at lower dropout rates and obtained similar results, though they become more pronounced as dropout increases.

*Table 8.* Summary of dropout schedules, correlation lengths, and test performance for MLPs on CIFAR-10, corresponding to Fig. 7.

| Schedule | $\xi_{\text{eff}}$ | $L/\xi_{\text{eff}}$ | Best Test Acc. (%) | Final Test Acc. (%) |
|---|---|---|---|---|
| No dropout | $\infty$ | 0.00 | $38.77 \pm 0.10$ | $38.34 \pm 0.12$ |
| Constant | 3.20 | 1.87 | $42.54 \pm 0.10$ | $41.84 \pm 0.10$ |
| Step (early) | 5.09 | 1.18 | $43.13 \pm 0.07$ | $42.67 \pm 0.09$ |
| Big step (1/3) | 6.67 | 0.90 | $43.29 \pm 0.08$ | $42.92 \pm 0.10$ |
| Double ($2\bar{h}$) | 2.54 | 2.36 | $41.46 \pm 0.08$ | $41.19 \pm 0.10$ |
| Triple ($3\bar{h}$) | 2.22 | 2.70 | $33.94 \pm 0.22$ | $33.82 \pm 0.23$ |

*Table 9.* Experimental and hyperparameter configuration for high overfitting regime. Corresponding performance metrics in Fig. 6.

| Parameter | Value / Description |
|---|---|
| *Network Architecture & Initialization* | |
| Network Width | 256 |
| Depth Multiplier | 2 |
| Weight Variance ($\sigma_w^2$) | 1.98 |
| Bias Variance ($\sigma_b^2$) | 0.02 |
| *Training Dynamics* | |
| Epochs | 50 |
| Batch Size | 75 |
| Learning Rate | $1 \times 10^{-4}$ |
| LR Schedule | Decay to minimum $1 \times 10^{-7}$ |
| Mean Dropout Field ($\bar{h}$) | 0.1 |
| Max Dropout Field ($h_{\max}$) | 0.2 (Step); 0.3 (Big step) |
| Step Active Fraction ($f$) | 1/2 (Step); 1/3 (Big step) |
| *General Setup* | |
| Number of Simulations | 20 |
| Dataset Usage | CIFAR-10 5,000 samples |
| Hardware | GPU A100 |

## D.4. Robustness Sweeps

To test whether scheduling is tied to one budget, width, or activation, we ran CIFAR-10 MLP sweeps (Figs. 8 and 9). The ReLU sweeps probe mean-field boundaries: at $\bar{h} = 0.15$, Step/Big step use nonperturbative local fields $0.30/0.45$; at $N = 64$, finite-width non-Gaussianity weakens the correlation-length prediction. The GELU MLP sweep gives the same best-accuracy ordering at $\bar{h} = 0.1$–constant $41.96 \pm 0.16$, Step $42.24 \pm 0.15$, Big step $42.46 \pm 0.12$–and Big step falls to $38.97 \pm 0.23$ at $\bar{h} = 0.15$, matching this perturbative-boundary picture. The final-epoch gains summarized in Table 3 use the last recorded epoch, giving $+2.04$ pp for the ReLU $\bar{h} = 0.1$ sweep and $+0.62$ pp for the GELU $\bar{h} = 0.1$ sweep.

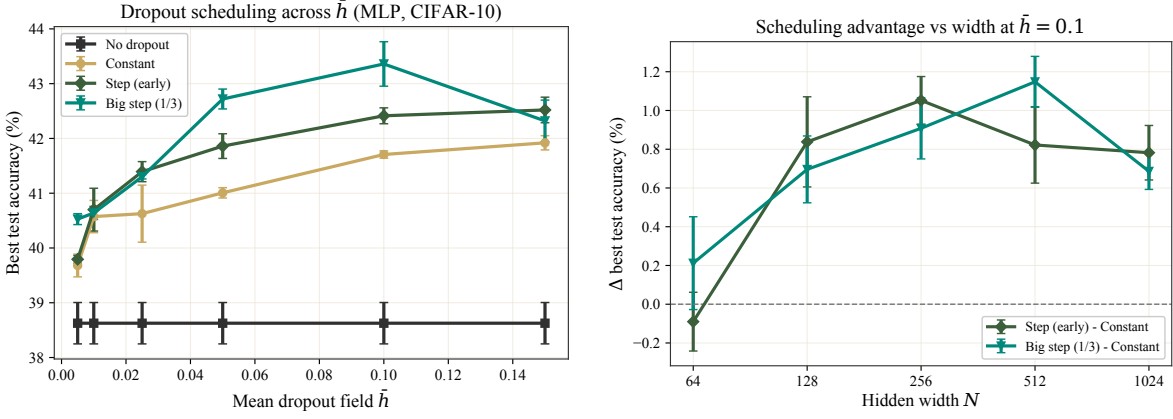

(a) Sweep over mean dropout field at width $N = 256$.

(b) Schedule advantage over constant dropout at $\bar{h} = 0.1$.

*Figure 8.* Robustness sweeps for depth-6 near-critical $\mathrm{ReLU}$ MLPs on CIFAR-10. Early schedules improve over constant dropout throughout the large-width regime, while $N = 64$ illustrates the expected finite-width boundary.

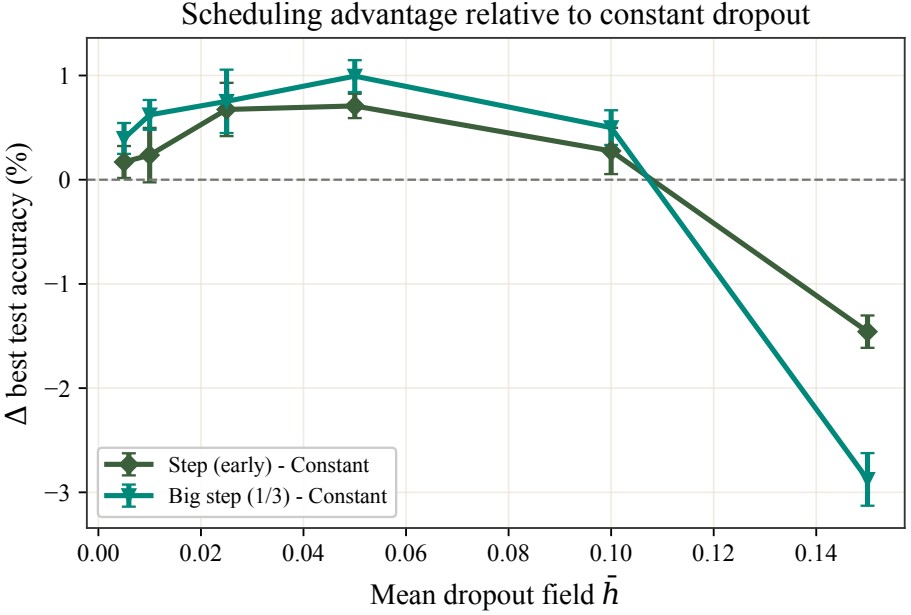

*Figure 9.* Smooth-activation $h$-sweep for depth-6 near-critical GELU MLPs on CIFAR-10 at width $N = 256$. Step-like schedules improve over constant dropout around $\bar{h} = 0.1$, while the largest field leaves the small-dropout regime where the mean-field perturbation is expected to be predictive.

## D.5. Transformers on CIFAR-100

Table 10 and Fig. 10 detail the architectural parameters and corresponding test dynamics for the CIFAR-100 Vision Transformer evaluation (Dosovitskiy et al., 2021; Krizhevsky, 2009); Table 11 reports the schedule comparison.

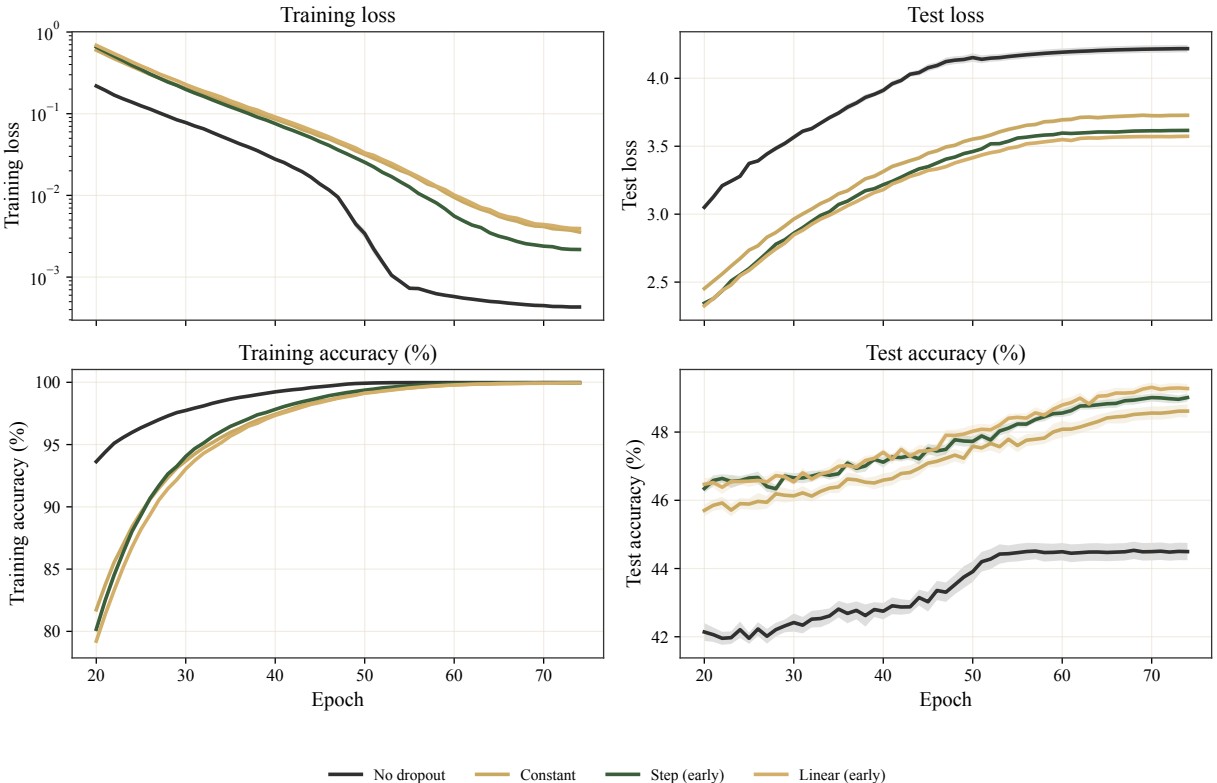

*Figure 10.* Finite-width Vision Transformer training and test curves at fixed mean dropout budget $\bar{h}$. We compare the no-dropout baseline, uniform dropout, decreasing linear ramps, and early step schedules. The cropped view shows epochs 20–75 for readability; the full curve is also in App. D.

Accuracy curves are shown in Fig. 10, with explicit results in Table 11. All reported errors are SEM. On CIFAR-100 (and similarly on CIFAR-10), concentrating dropout earlier consistently improved accuracy and reduced overfitting. Interestingly, linear decreasing schedules, which put more dropout in early layers, often performed especially well in transformers; residual connections were introduced in (He et al., 2016), and mean-field analyses show that they ease information propagation (Yang & Schoenholz, 2017). We suspect this makes distributing regularization early preferable empirically, though both early weighted and step-like schedules outperformed the constant baseline.

We focus on controlled regimes rather than ImageNet-scale benchmarking because our central claims are mechanistic and derived in a mean-field initialization framework, which is most cleanly tested when finite-width corrections remain subdominant. The experimental validation is designed to test the theory in settings where MFT predictions apply cleanly, rather than to establish state-of-the-art performance on large-scale datasets such as ImageNet (Deng et al., 2009). ImageNet-scale training introduces substantial confounders from augmentation and extensive hyperparameter tuning that can obscure MFT effects; we leave that evaluation as future work. In these controlled runs, the predicted schedules outperform constant dropout, illustrating the practical utility of the mean-field framework.

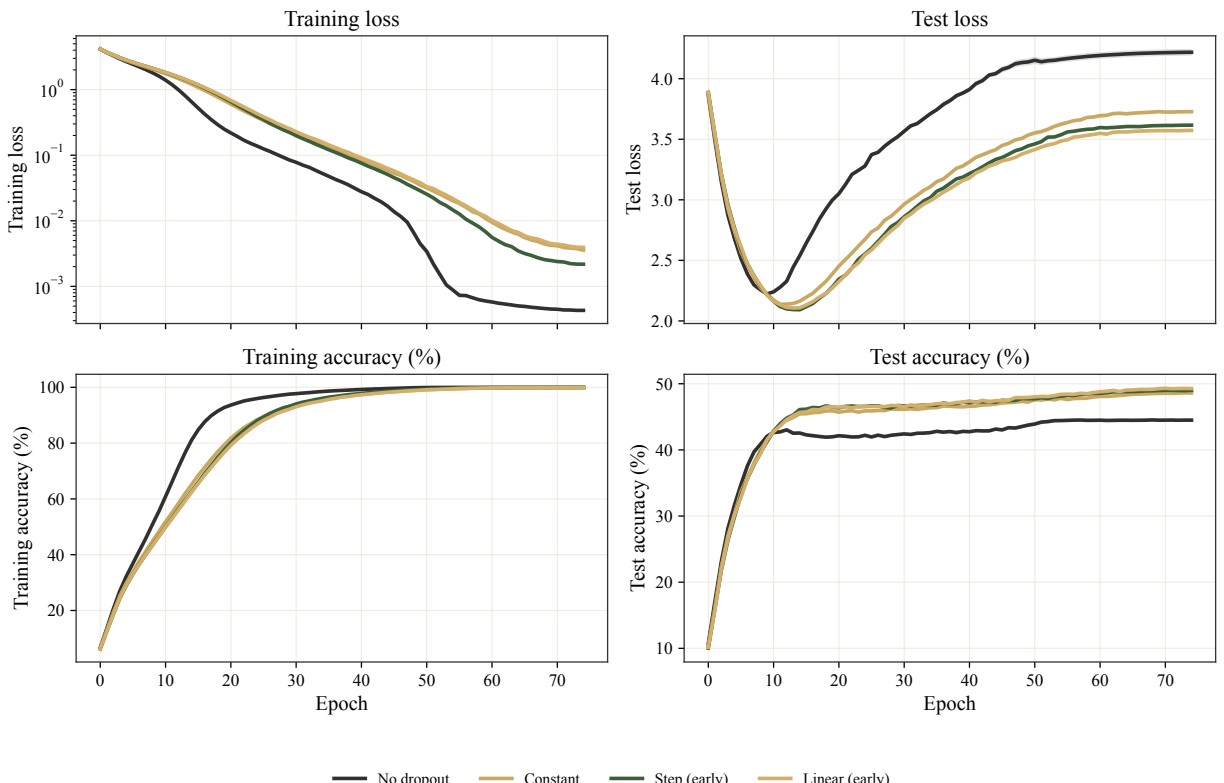

*Figure 11.* Extended training curves for Figure 10, showing the full 75 epochs.

*Table 10.* Experimental configuration for Vision Transformer (ViT) simulations.

| Parameter | Value / Description |
| --- | --- |
| *ViT Architecture* | |
| Embedding Dimension | 128 |
| Depth | 12 |
| Number of Heads | 16 |
| MLP Ratio | 4.0 |
| Patch Size | 4 |
| *Training Dynamics* | |
| Epochs | 75 |
| Batch Size | 250 |
| Learning Rate | $5 \times 10^{-4}$ |
| LR Minimum | $5 \times 10^{-6}$ |
| Weight Decay | 0.00 |
| Mean Dropout Field ($\bar{h}$) | 0.1 |
| Max Dropout Field ($h_{\max}$) | 0.2 |
| Schedules | None, Constant, Step (early), Linear (decreasing) |
| *General Setup* | |
| Number of Simulations | 10 |
| Dataset Usage | Full CIFAR-100 |
| Evaluation Frequency | Every 1 Epoch |
| Hardware | GPU A100 |

*Table 11.* Test accuracy across different dropout schedules for ViT architecture on CIFAR-100, corresponding to Fig. 10.

| Schedule | Best Test Acc | Final Test Acc |
|---|---|---|
| No dropout | $44.61 \pm 0.25$ | $44.50 \pm 0.25$ |
| Constant | $48.69 \pm 0.18$ | $48.61 \pm 0.20$ |
| Step (early) | $49.10 \pm 0.11$ | $49.01 \pm 0.12$ |
| Linear (decreasing) | $49.38 \pm 0.14$ | $49.27 \pm 0.14$ |

### D.6. Transformer Ablations

To isolate the effect of dropout scheduling on distinct transformer components, we performed component-level ablations. We tested transformers with dropout applied exclusively to the attention block, the MLP block, or both, comparing step-like and constant schedules against a no-dropout baseline. Experimental hyperparameters are listed in Table 12; they largely mirror the main transformer experiment, though we conducted five simulations per configuration. We repeated the ablation on CIFAR-10.

*Figure 12.* Component ablations applying dropout schedules to the attention block, MLP block, or both. The figure compares no dropout, constant dropout, and step-like dropout.

*Table 12.* Experimental configuration for Vision Transformer (ViT) ablation study.

| Parameter | Value / Description |
| --- | --- |
| *ViT Architecture* | |
| Embedding Dimension | 128 |
| Depth | 12 |
| Number of Heads | 16 |
| MLP Ratio | 4.0 |
| Patch Size | 4 |
| *Training Dynamics* | |
| Epochs | 75 |
| Batch Size | 1000 |
| Learning Rate | $5 \times 10^{-4}$ |
| LR Minimum | $5 \times 10^{-6}$ |
| Weight Decay | 0.00 |
| Mean Dropout Field ($\bar{h}$) | 0.1 |
| Max Dropout Field ($h_{\max}$) | 0.2 |
| Schedules | None, Constant, Step (early) |
| Ablation Modes | Both, Attention-only, MLP-only |
| *General Setup* | |
| Number of Simulations | 5 |
| Dataset Usage | Full CIFAR-10 |
| Evaluation Frequency | Every 1 Epoch |
| Hardware | GPU A100 |

*Table 13.* Component-level ablations on CIFAR-10, comparing Constant and Step-like dropout schedules applied exclusively to the MLP block, Attention block, or both. The Step (early) schedule consistently outperforms the Constant schedule across all configurations.

| Configuration | Best Test Acc | Final Test Acc |
|---|---|---|
| No dropout | $72.22 \pm 0.31$ | $72.15 \pm 0.33$ |
| MLP only (Constant) | $72.96 \pm 0.26$ | $72.88 \pm 0.28$ |
| MLP only (Step early) | $73.38 \pm 0.56$ | $73.29 \pm 0.55$ |
| Attn only (Constant) | $74.00 \pm 0.20$ | $73.95 \pm 0.21$ |
| Attn only (Step early) | $74.48 \pm 0.20$ | $74.41 \pm 0.19$ |
| Both (Constant) | $74.92 \pm 0.22$ | $74.78 \pm 0.23$ |
| Both (Step early) | $75.47 \pm 0.27$ | $75.30 \pm 0.29$ |

