# OpenReview forum: "Dropout Universality: Scaling Laws and Optimal Scheduling at the Edge-of-Chaos"
_ICML.cc/2026/Conference — ICML 2026 regular_

### Official Review · Reviewer_GSTB · 2026-03-10

**Soundness:** 2
**Presentation:** 2
**Significance:** 2
**Originality:** 2
**Overall Recommendation:** 4
**Confidence:** 5

**Summary:**

This manuscript employs mean-field theoretical frameworks to investigate how dropout influences critical signal propagation at the edge-of-chaos regime. The authors conceptualize dropout as a correlated "external field" perturbation that disrupts the perfect-alignment fixed point, thereby establishing a Landau-type equation for the decorrelation order parameter. They subsequently analyze distinct critical scaling behaviors under two categories of activation functions: smooth and kinked. Building upon this theoretical foundation, the paper introduces a two-parameter scaling collapse and derives scheduling principles under fixed dropout budgets, advocating for step-like, front-loaded dropout allocation strategies. The empirical component validates several theoretical phenomena on MLPs and Vision Transformers, reporting performance gains from early-concentrated dropout relative to constant dropout regimes.

**Compliance With Llm Reviewing Policy:**

Affirmed.

**Final Justification:**

The authors partially addressed the concerns I raised in Question 1, but I believe the issue in Question 2 still remains. Therefore, I will increase my score by only one point.

**Key Questions For Authors:**

See Weekness.

**Limitations:**

Yes.

**Strengths And Weaknesses:**

Strengths
1. The theoretical contribution exhibits notable originality with rigorous derivations. Rather than merely discussing empirical effects of dropout, the authors systematically integrate dropout into the edge-of-chaos/mean-field paradigm, analyzing how it modulates correlation propagation and induces divergent critical exponents between smooth and kinked activation functions. This theoretical apparatus is further extended to inform dropout scheduling design principles.

Weaknesses
1. The introduction directly asserts a "proof" that smooth and kinked activations belong to distinct universality classes. However, the core derivations primarily rest upon MLP mean-field approximations, local expansions, and analysis of non-analytic terms in the kinked scenario. While this constitutes compelling analytical evidence, whether it constitutes rigorous "proof" for broader activation function families remains debatable. Similarly, conclusions regarding Transformers rely more heavily on empirical corroboration and plausible extrapolation, as the authors explicitly acknowledge that primary derivations were conducted within the MLP framework, with applicability to Transformers depending substantially on the assertion that "activation analytic structure governs scaling behavior."
2. The experimental validation leans toward theoretical corroboration rather than robust methodological validation, thus limiting empirical persuasiveness. The authors explicitly state experiments were conducted under controlled conditions to prevent finite-width corrections and data augmentation from obscuring signal propagation effects, rather than pursuing benchmark performance—consistent with the paper's positioning, yet implying limited experimental scope, task complexity, and magnitude of methodological gains. Particularly regarding scheduling, while core conclusions receive support from MLP and ViT experiments, overall improvement margins remain modest. Moreover, the critical tie-breaker explaining "why front-loaded steps outperform back-loaded ones" primarily invokes an auxiliary rank-collapse mechanism—a combination of reasonable motivation and empirical support that has not yet achieved the same rigor as the preceding mean-field derivations.

---

> ### Author Rebuttal · Authors · 2026-03-31
>
> We thank the reviewer for their careful reading and constructive feedback. We address both weaknesses in order.
>
> We agree that the manuscript should more carefully distinguish between what is formally derived and what is empirically extrapolated; we will replace "proof" with "analytical evidence" wherever the claim extends beyond the MLP setting. That said, we wish to clarify that the analytical evidence for two universality classes is considerably stronger than a pure MLP result. The smooth/kinked distinction is a consequence of the analytic regularity of the activation at the level of the Gaussian correlation channel. As shown in Appendix~E, for any activation with an ordinary kink, the near-alignment expansion develops a branch point at $c=1$ with exponent $3/2$ fixed by the geometry of the Gaussian tube around $u_1 = u_2$. Crucially, this mechanism is local: it depends only on the analytic structure of $F(c)$ near $c=1$, which is determined by the activation and the Gaussian measure, not the architecture. In CNNs, the recursion becomes $\Sigma^{\ell+1} = A \star \mathcal{C}(\Sigma^\ell)$ (Xiao et al., 2018), where $\mathcal{C}$ is the same Gaussian kernel controlling local correlation dynamics and $A$ is the channel-averaging operator induced by the convolutional filter; on the translation-invariant sector, $A$ acts trivially and the CNN map collapses to the scalar MLP recursion, so the smooth/kinked local structure is preserved. In ResNets, Yang and Schoenholz (2017) show that skip connections change convergence from exponential to polynomial, but the residual branch still passes through $\mathcal{C}(q,c)$, leaving the local analytic structure intact. We will add concise versions of the CNN and ResNet arguments to the camera-ready draft.
>
> Regarding experimental validation: we agree the scope is limited, and this is by design. The goal of this exercise was to test whether the theoretical predictions hold in a setting where confounding factors are stripped away. We hope the preceding discussion, arguing that a solid understanding of the base case extends naturally to more complex architectures, will convince the reviewer that the experiments are appropriate for the scope of our paper. We also note that while absolute margins are modest, they are statistically significant and achieved at zero additional computational cost, requiring as little as a single-line script modification.
>
> Regarding the rank-collapse argument: this is explicitly presented as a complementary heuristic, and the core scheduling results do not depend on it. We view rank collapse as analogous to selecting the physically sensible branch of solutions after solving a differential equation, a point where the practitioner's judgment must be applied. Rank collapse and $c^* \to 1$ are related but non-equivalent diagnostics; the former requires tracking higher moments beyond the two-point function. A quantitative joint optimization incorporating feature diversity alongside $\xi_{\mathrm{eff}}$ is a question deserving its own paper. We emphasize, however, that our current treatment is already predictive: in our experiments, schedules with longer $\xi_{\mathrm{eff}}$ consistently generalize better. If space permits, we will also include a simple depth-dependent rank proxy to make the tiebreaker argument more concrete.

---

> > ### Author Rebuttal · Reviewer_GSTB · 2026-04-02
> >
> > The authors addressed my concerns well, and I will raise my score.

---

> > > ### Author Response · Authors · 2026-04-08
> > >
> > > We are thankful to the reviewer for engaging with our work and raising their score, and we are pleased that our arguments resolved their concerns.

---

### Official Review · Reviewer_7Rde · 2026-03-11

**Soundness:** 3
**Presentation:** 3
**Significance:** 3
**Originality:** 3
**Overall Recommendation:** 4
**Confidence:** 3

**Summary:**

The manuscript investigates the behavior of randomly initialized deep neural networks under dropout through the lens of Mean-Field Theory. A central theoretical contribution is the classification of activation functions into distinct universality classes. These classes are shown to have different critical exponents. Furthermore, the authors use this framework to frame optimal dropout scheduling as a variational problem. Experiments on MLPs and ViTs on the CIFAR-10 and CIFAR-100 datasets verify the effectiveness of the optimal scheduling.

**Compliance With Llm Reviewing Policy:**

Affirmed.

**Final Justification:**

This is a technically solid paper. I recommend weak acceptance due to the remaining theory-practice gap.

**Key Questions For Authors:**

1. Does the distinction between kinked and smooth activations imply that one class is inherently more robust or preferred in practice when applying dropout near the edge of chaos?

2. In Appendix F, the paper analyzes different activations using Hermite decompositions. However, this spectral perspective does not seem explicitly utilized in Appendix C when deriving the critical exponents. Could you clarify the exact mathematical relationship between the Hermite polynomial decay rates and the emergence of different critical exponents?

3. The rank-flow tie-breaking argument is logically sound, but it remains purely theoretical in the text. Could the authors provide an empirical plot?

4. The authors note that their methods are "orthogonal" to methods like Curriculum Dropout. Even if orthogonal, a direct empirical comparison against these existing dropout variants would be highly beneficial to demonstrate the state-of-the-art utility of the derived schedules. For example, does a combination of the optimal curriculum and an existing dropout variant gives a better result?

**Limitations:**

Yes

**Strengths And Weaknesses:**

Strength: Formulating dropout in the mean field framework and the identification of distinct universality classes based on the analytic structure of activation functions is novel and a significant contribution to the field. The authors also translate this framework into a concrete, actionable algorithm, which is quite appreciated.

Weakness: 1. The paper establishes that smooth and kinked activations belong to distinct universality classes governed by different critical exponents. However, the practical implications of this distinction remain abstract, e.g., how this should influence a practitioner's choice.

2. The paper lacks direct empirical verification of the mechanism behind the optimal scheduling (e.g., tracking the effective rank proxy across layers during training) and comparison against established dropout scheduling methods.

---

> ### Author Rebuttal · Authors · 2026-03-31
>
> We thank the reviewer for the detailed feedback. We address each point below and indicate the concrete changes we will make in the revision.
>
> We would also like to kindly point out two additional contributions which we believe are noteworthy and that we may further emphasize in the revised version. First, the framework of promoting hyperparameters to depth-dependent fields and solving variational problems over them is, to our knowledge, new to the MFT community. We demonstrate it here for dropout scheduling, but the same machinery applies immediately to depth-dependent learning rates, residual-branch strengths, or any layer-wise hyperparameter that enters the mean-field recursion; e.g. (Yang et al., 2017) show that optimal initialization cannot be depth-independent, excluding He/Xavier as globally optimal, and our formalism provides the natural language for posing and solving such problems. We see the dropout application as the first worked example of a broader program. Second, the fixed point is not destroyed by dropout but perturbatively displaced, so all mean-field quantities (correlation lengths, depth scales) remain well-defined, something missed in (Schoenholz et al., 2017) and that allows for analytical control in extended regimes.
>
> Regarding the practical implications of the two universality classes (Weakness~1 and Q1): in our opinion, the practical implication is best understood as a tradeoff. Smooth activations better preserve $\xi_c$ at fixed small dropout field through $\xi_c \sim h^{-1/2}$, whereas kinked activations are more forgiving to imperfect tuning because their crossover window is wider. Our claim is not that one class is uniformly "better," but rather that the same dropout field changes the distinguishable depth and schedule sensitivity differently for smooth and kinked activations. For ReLU specifically, the zero-bias He initialization sits exactly at criticality, which is one reason ReLU-type networks often perform well. Conversely, our work helps explain why smooth variants such as GeLU often perform even better: they suffer less degradation from dropout precisely because their correlation length decays more gently than ReLU ($h^{-1/2}$ vs.\ $h^{-1/3}$). We will add a practitioner-oriented paragraph in the revision that translates these scaling laws into concrete schedule-design guidance.
>
> On the mathematical relationship between Hermite decay and the critical exponents (Q2): we agree this bridge should be stated much more explicitly. The logical chain is as follows. Smooth activations have fast Hermite decay, so $F(c)$ is analytic as $c \to 1$, yielding an ordinary Landau expansion. Kinked activations have power-law Hermite tails, so high-degree modes contribute collectively, producing a branch-point nonanalyticity. Appendix G diagnoses when the nonanalyticity is present through the Hermite decomposition, while Appendix C translates that structure into the critical exponents. As the paper is already mathematically dense, we deliberately kept the appendices self-contained to avoid forcing the reader to read the appendix to understand the main body. We will add an explicit bridge paragraph connecting these two appendices so the reader does not need to reconstruct this chain independently.
>
> On the rank-flow empirical plot (Q3): we will include a plot tracking effective rank during training to provide direct mechanistic evidence. We note that rank collapse (representations converging to a low-dimensional subspace) and $c^{\ast} \to 1$ are related but nonequivalent diagnostics: the former requires tracking higher moments, not just the two-point function. A quantitative joint optimization incorporating feature diversity alongside $\xi_{\text{eff}}$ is a rich question that we believe deserves its own dedicated study.
>
> On the relation to Curriculum Dropout (Q4): we agree that a direct empirical comparison would add value, and we will include one in the revision along with a discussion of composability. The comparison will make this orthogonality explicit and test whether the combination yields additive gains.
>
> In summary, the revision will (i) add an operational recipe turning the scaling laws into schedule-design guidance for practitioners; (ii) add an explicit bridge paragraph connecting the Hermite analysis to the analytic vs. nonanalytic structure of the correlation map in Appendix~C, and hence to the distinct exponents; (iii) add a layer-resolved effective-rank plot during training to provide direct mechanistic evidence; and (iv) include a direct comparison with Curriculum Dropout and clarify the distinction between our spatial scheduling framework and temporal dropout methods, including composability. We will also expand the practical discussion of robustness near criticality, especially for ReLU-type activations. We hope this addresses the reviewer's concerns.

---

> > ### Author Rebuttal · Reviewer_7Rde · 2026-04-02
> >
> > Thank you for the detailed rebuttal.
> >
> > While I appreciate the conceptual explanation linking Appendix F to the Appendix C, all derivations in Appendix C are not based on the results of Appendix F. That's why I do not understand the relation between critical exponents and the Hermit expansion.
> >
> > Your hypothesis regarding GeLU outperforming ReLU due to a gentler correlation length decay is an interesting takeaway. However, this is a strong claim about practical performance that requires empirical validation in the paper.
> >
> > Finally, the absence of empirical results concerning rank flow comparisons with other dropout schedules makes it difficult to fully justify the paper at this stage.
> >
> > Given the remaining gap, I will maintain my score of Weak Accept.

---

> > > ### Author Response · Authors · 2026-04-08
> > >
> > > We thank the reviewer for the careful follow-up and address the three remaining points.
> > >
> > > On the relation between Appendices F and C, we would like to offer the sequence of calculations that led formed the basis of much of this paper, to hopefully shine light on the connection between these sections and how the Hermite decomposition and variational problem outlined in Appendix F were the smoking gun that led us to consider the possibility of different universality classes. The investigation began by trying to solve for the activation functions that would lead to the highest $\xi_c$. After realizing this led to Hermite polynomials as a natural basis, we were interested to learn how different activations are decomposed in this basis to find a simpler equation for the correlation recursions in special cases. It was then that we found that the spectral decompositions of tanh and ReLU were qualitatively different, as one decayed exponentially while the other followed a power law. In the Hermite basis the correlation map reads
> > > \begin{equation}
> > >     F(c) \;=\; \frac{\sigma_w^2 \sum_{n \geq 0} a_n^2\, c^n + \sigma_b^2}{q^\ast},
> > > \end{equation}
> > > so the analytic structure of $F(c)$ near $c = 1$ is controlled entirely by the decay rate of the coefficients $a_n$. Rapid decay, typical of smooth activations, keeps all derivatives at $c = 1$ finite, so $F$ is analytic, the ordinary Landau expansion applies, and one lands in the smooth branch of Appendix C with its associated exponents. A slow power-law tail, typical of kinked activations, means that high Hermite degrees contribute collectively as $c \to 1$, the Taylor expansion is no longer the correct organizing principle, and one picks up the branch point that produces the $m^{3/2}$ term and the kinked exponents. $F(c)$ is the object that connects both appendices: Appendix F diagnoses its analytic class, and Appendix C computes the Landau theory of that class. We will add a short paragraph stating this explicitly so the reader does not have to reconstruct the logical dependence.
> > >
> > > On the GeLU versus ReLU remark, we agree that the original wording was too strong. We did not intend a standalone empirical performance claim, but rather a suggestive reading from a MFT lens, and we will soften the sentence accordingly in the revision.
> > >
> > > On the rank-flow comparisons, we acknowledge the gap. Rank flow is presented in the paper as a plausible tie-breaking mechanism beyond mean field, and we agree that a dedicated empirical study comparing rank dynamics across dropout schedules would strengthen this part of the argument. We will add the schedule comparisons that are most directly relevant to this point in the revision, while being careful not to overclaim what they establish about the mechanism itself.
> > >
> > > We appreciate the careful reading and the constructive engagement throughout the discussion.

---

### Official Review · Reviewer_rHNj · 2026-03-11

**Soundness:** 3
**Presentation:** 2
**Significance:** 2
**Originality:** 3
**Overall Recommendation:** 3
**Confidence:** 3

**Summary:**

The authors use mean-field theory to develop a theoretical description of dropout in large, overparameterized deep feedforward networks. They introduce an order parameter that captures residual correlations, along with two couplings corresponding to a linear stability measure and the transmission of correlations away from perfect alignment. In terms of this order parameter and the two couplings, they derive an equation reminiscent of the Landau equation of state.
Using this framework, they show that smooth activation functions and ReLU activations belong to different universality classes, characterized by their critical exponents. Furthermore, the authors derive dropout scheduling rules that maximize the correlation length of samples under the assumption of a fixed dropout budget. Finally, they demonstrate that, under this assumption, the proposed schedule outperforms constant dropout when applied to the early layers of the network.

**Compliance With Llm Reviewing Policy:**

Affirmed.

**Key Questions For Authors:**

1.
The derived schedules appear to work primarily for early layers, as also the authors argue in connection with rank collapse.
This suggests a trade-off between increasing correlation length and maintaining a sufficiently large effective rank.
Can your theory capture this trade-off quantitatively?
It seems to be discussed mainly at a qualitative level.
It would be helpful to better understand when the theoretically optimal schedule can be applied in practice, and under what conditions it breaks down due to rank collapse.

2.
Is there a qualitative difference between dropout noise and other forms of regularization? After line 418, the paper states that dropout increases the effective rank of the second moments.
However, similar effects seem to arise from other regularization mechanisms, such as additive noise applied at each layer.
Specifically Eq. (49) appears similar to adding uncorrelated noise across samples, which would lead to a diagonal contribution to the second-moment matrix.
Could the authors clarify what distinguishes dropout from these other forms of noise within the proposed framework?


3.
The authors should note that, although their theory predicts an optimal dropout schedule, it only outperforms constant dropout when applied to early layers (front-loaded).
In practice, other schedules (such as the 'Linear (decreasing)' schedule in Figure 5) can perform better, suggesting interactions with rank collapse that are not fully captured by the current theory. A quantitative characterization of the trade-off between maximizing correlation length and avoiding rank collapse would strengthen the work.

**Limitations:**

Yes.

**Strengths And Weaknesses:**

Strengths:

The paper provides a clearly written introduction.
The theoretical setup are explained well and are easy to follow.
The conceptual idea is well formulated.
The results are relevant for modern architectures like Transformers.
The appendix is detailed and provides additional derivations and explanations.
The paper connects between statistical physics and machine learning.
The distinction between qualitative behaviors of different activation functions is interesting and potentially insightful.


Weaknesses:

While the use of Landau theory is conceptually appealing, it is not entirely clear that it provides substantial additional insights beyond what can already be obtained from the underlying mean-field analysis.
The derivation of the Landau equation stems from a Taylor expansion of the correlation transmission function, raising the question of how much new conceptual understanding the Landau framework itself contributes.
The introduction of the order parameter and the couplings appears somewhat ad hoc. They seem to be introduced primarily to enable the Landau-type formulation.
The differences in critical exponents between activation functions appear to arise from the behavior of the derivatives of the correlation transmission function at the edge of chaos. As a result, the universality-class interpretation may be somewhat limited in terms of additional explanatory power.

Presentation:

Figures 1 and 2 are not referenced in the main text. Instead, the paper frequently refers to material in the appendix. It would improve readability if the main text referenced and discussed the figures directly.
The paper promises a comparison with Multi-Layer Perceptrons and Vision Transformers in the abstract, but this comparison only appears in the appendix. It would be beneficial to include it in the main text instead of some minor equations.
Some mathematical symbols are introduced without clear definition or explanation. For example:

The quantity Delta in Eq. (35)
The parameter lambda in Eq. (37)
The 'effective rank proxy' introduced in Eq. (45)


Significance:
Dropout is a widely used regularization method in deep learning, and therefore developing a mean-field theoretical understanding of it is relevant.

Originality*
The paper introduces a heavy formalism, but much of the meaning of the results appears somewhat obscured by it and the large collection of equations.
For example, e.g. Eq. (21) essentially states that the noise induces a residual correlation, but its meaning is hidden by the introduction of the new variable names.

---

> ### Author Rebuttal · Authors · 2026-03-30
>
> We thank the reviewer for their detailed feedback and address each point below.
>
> Regarding the concern that the Landau framing may not provide substantial insight beyond the Taylor expansion of the correlation map: we respectfully disagree. The Landau language buys three things the raw expansion does not. First, the scaling collapse (Sec. 3.2) requires identifying rescaled variables ($\tilde{m}$, $\tilde{t}$) and showing all curves fall on a single universal function (Figs. 2 and 4); this is not visible from the Taylor coefficients alone. Second, the variational scheduling relies on promoting the dropout field $h$ to a depth-dependent quantity $h\_\ell$ and minimizing an effective inverse correlation length; this idea (spatially varying fields) does not follow from a fixed-depth Taylor expansion. The same machinery extends to depth-dependent learning rates, residual-branch strengths, or any layer-wise hyperparameter; e.g. (Yang et al., 2017) show that optimal initialization cannot be depth-independent. Dropout is simply the first chapter in a broader program aiming to gain a first principles understanding of what the best scheduling profiles are for different common parameters in modern architectures. Third, the universality classification (smooth vs. kinked, Table 2) explains a previously unexplained observation: (Yang et al., 2017) noted qualitatively different tanh vs. ReLU asymptotics but had no framework to explain why, which our work does. While we only derived the mean field theory treatment in MLPs, we expect a similar qualitative behavior for ResNets, as the Gaussian Kernel is inherited, though the information degradation is mitigated by the skip-connections, and the MFT treatment is more subtle.
>
> More generally, universality class behavior is extensively studied in statistical physics, and the scaling collapse itself is a tool borrowed directly from that literature; once a system is identified as belonging to a given class, the full suite of results (exponents, crossover functions, finite-size scaling) carries over without additional derivation. The variables $(m, t, h)$ are not ad-hoc; they are the standard Landau variables for any system with a displaced fixed point. We also note that the fixed point is not destroyed by dropout but perturbatively displaced, so all mean-field quantities remain well-defined, something missed in (Schoenholz et al., 2017); this allows us to retain analytic control in this regime.
>
> On presentation: we agree that Figures 1 and 2 should be referenced in the main text and will fix this, along with a clearer reference to the MLP/ViT comparison in the main text. We will add explicit definitions or back-references for $\Delta(\rho\_\ell)$ (Eq. 35), $\lambda$ (Eq. 37), and the rank proxy (Eq. 45). We appreciate these specific pointers.
>
> The trade-off between correlation length and rank collapse is a delicate and subtle issue: rank collapse (representations converging to a low-dimensional subspace) and $c^\ast \to 1$ (inter-input correlations converging to unity) are related but nonequivalent. Our framework addresses the latter; the former requires tracking higher moments, not just the two-point function. A quantitative joint optimization incorporating feature diversity alongside $\xi\_{\text{eff}}$ is a question deserving its own paper. We, however, emphasize that our current treatment is already predictive: in our experiments, schedules with longer $\xi\_{\text{eff}}$ consistently generalize better.
>
> The qualitative distinction between regularization and other noise regularizers is that dropout is multiplicative, not additive. Additive noise shifts the variance by a signal-independent constant and leaves the correlation map at $c=1$ unaffected, since both inputs receive the same additive structure. Dropout acts as a symmetry-breaking field $h > 0$ for alignment, whereas additive noise preserves the $c=1$ fixed point entirely.
>
> Finally, the reviewer notes that Linear (decreasing) schedules can outperform step schedules. In the MLP experiment (Fig. 5, Table 5) the ranking tracks $\xi\_{\text{eff}}$ monotonically, exactly as predicted. The transformer results reveal an additional effect: residual connections mitigate correlation collapse from exponential to polynomial in depth (Yang et al., 2017), so the architecture itself partially protects information propagation as previously discussed. With that bottleneck relaxed, having nonzero regularization everywhere becomes more important than the marginal gain in correlation length. This is why our early step function schedule outperforms the baseline in our ViT experiments, and why schedules with broader coverage (e.g. linear schedules) can compete with step schedules that leave half the network unregularized.
>
> We will add a dedicated limitations section and address the presentation issues raised above.

---

> > ### Author Rebuttal · Reviewer_rHNj · 2026-04-03
> >
> > I have follow-up questions for the authors:
> >
> > I thank the authors for responding to my concerns.
> >
> > 1.
> > Thank you again for the explanation regarding universality classes. I trust your analysis and calculations showing the universality classes, and I appreciate the clarification of their meaning, as well as the pointers to the literature.
> >
> > My original confusion regarding this Landau-type formulation stems from the notion that such a framework typically involves a Landau ('free energy') functional, whose minimum corresponds to an equilibrium state.
> >
> > It is not clear to me what the Landau functional is in your formulation, nor how (or whether) it can be meaningfully interpreted.
> > While it is clear to me what the order parameter (m=1-c^*) (magnetization) represents, it is not clear to me whether it assuming its equilibrium value (minimizing the functional) can be interpreted in the context of neural networks.
> >
> > 2.
> > The motivation behind my question regarding front-loading is the following: it is somewhat unsatisfying that the explanation for the optimal dropout schedule relies on an additional rank-collapse intuition, which does not appear to be as well justified as the mean-field calculations underlying the optimal dropout schedule.
> >
> > 3.
> > In their rebuttal, the authors state that ''Additive noise shifts [...] leaves the correlation map at (c = 1) unaffected, [...].''. However, this appears to hold only if different samples receive the same noise realization, correct?
> > If, instead, the additive noise realization differs across samples (as is also the case for the dropout mask (as described on page 3 of the manuscript)) then the correlation map at (c = 1) would indeed be affected. This raises the question of whether additive noise and dropout noise are, in fact, as different as suggested.
> >
> > Thank you for taking the time to address my questions.
> > If my concerns are satisfactorily resolved, I would be happy to revise my score accordingly.
> >
> >
> > Score: maintained

---

> > > ### Author Response · Authors · 2026-04-08
> > >
> > > We thank the reviewer again for the careful follow-up. These are subtle points, and we agree the current draft should make the logical structure more explicit.
> > >
> > > On the Landau functional. The relevant equilibrium here is not thermodynamic equilibrium of a trained network, but the stable fixed point of the forward correlation recursion. Dropout perturbatively shifts the fixed point from $c^\star=1$ to $c^\star(\rho)$, with order parameter is $m^\star=1-c^\star(\rho)$. The Landau functional is therefore a local auxiliary potential, not derived from a partition function, but instead constructed so that its stationarity condition reproduces the equation of state (EOS) we already have from the recursion, with $f''(m^\star)>0$ coinciding with dynamical stability of the corresponding fixed point under depth iteration. This reverses the usual logic, in which one differentiates a given potential to obtain an EOS. The construction is analogous in spirit to the 1PI effective action $\Gamma$ in quantum field theory, which is defined so that its stationarity condition reproduces the quantum equations of motion, with convexity of $\Gamma$ encoding stability of the corresponding state, mirroring $f''(m^\star) > 0$.
> > > Near perfect alignment, (see App.C)
> > > \begin{equation}
> > > h+t\,m-\frac{g_\rho}{2}m^2=0,
> > > \end{equation}
> > > equivalently
> > > \begin{equation}
> > > \frac{\partial f}{\partial m}=0,\quad
> > > f(t,m;h)=-\frac{t}{2}m^2+\frac{g_\rho}{6}m^3-hm.
> > > \end{equation}
> > > Its meaning in the neural-network setting is that the local minimum of $f$ coincides with the stable branch reached under depth iteration. On the physical branch, continuously connected to $c^\star\to1$ as $h\to0$,
> > > \begin{equation}
> > > f''(m^\star)=-t+g_\rho m^\star>0,
> > > \end{equation}
> > > so the stationary point is a local minimum exactly when the corresponding fixed point of the recursion is dynamically stable; the other root has $f''<0$ and is the unstable branch. We will make this interpretation explicit in the revision and, if the reviewer prefers, refer to $f$ as a pseudo-free energy.
> > >
> > > On the rank collapse. We agree the present draft did not separate sharply enough what is established by mean-field analysis from what is imposed to achieve better performance as opposed to simply maximizing $\xi_c$. We will revise the text to make that distinction explicit. Crucially, MFT alone already determines the optimal class of schedules, and rank flow enters only to resolve a residual degeneracy that mean-field analysis is blind to; front-loading is not needed to derive the optimal schedule at the mean-field level. Because $\xi_{\mathrm{eff}}$ depends on the profile $\{h_\ell\}$ only through symmetric sums, all permutations of a given step profile lead to exactly the same extremal$\xi_c$, but not all will improve performance, which is ultimately what we want. The role of rank flow is then to resolve this residual degeneracy; it is not an additional assumption needed for the variational result itself, and would be unnecessary if we only cared about information propagation. The intuitive reason why performance improves when front-loading dropout is that this choice is inherently preventative: once the representation has already collapsed toward a low-rank subspace, later dropout mainly inflates the diagonal and cannot reconstruct lost modes. Therefore, among the step-like profiles that are exactly degenerate at the level of $\xi_{\mathrm{eff}}$, the best choice is to place the active block early, where it can still prevent collapse rather than act after the fact. We hope this addresses the reviewer's concern.
> > >
> > > On additive versus multiplicative noise. The reviewer is right that our previous wording overstated the distinction. Independent sample-wise additive noise also shifts the $c=1$ fixed point: it is easy to see that if two signals have variance $q$ and one adds independent zero-mean noise of variance $\sigma_\eta^2$ to each, then
> > > \begin{equation}
> > > c'=\frac{q}{q+\sigma_\eta^2}<1.
> > > \end{equation}
> > > So loss of perfect alignment is not unique to dropout, and the same Landau equation of state applies with a different microscopic field $h(\sigma_\eta^2)$. We view this as broadening, rather than weakening, the scope of the framework: the universality-class argument covers a family of independent alignment-breaking deformations rather than dropout specifically. What is special about dropout is instead structural: it induces the signal-dependent deformation through the Hadamard product, which is what ties it naturally to the second-moment shrinkage identity used in the rank-flow argument.
> > >
> > > In summary, we will make the Landau-functional interpretation explicit, clarify that MFT fixes the optimal dropout profile while rank flow only resolves the residual degeneracy, and broaden the discussion to alignment-breaking regularizers more generally. We hope these clarifications resolve the remaining conceptual concerns, and we would be grateful if the reviewer would consider revising the score accordingly.

---

### Official Review · Reviewer_uFjt · 2026-03-12

**Soundness:** 3
**Presentation:** 4
**Significance:** 2
**Originality:** 3
**Overall Recommendation:** 4
**Confidence:** 5

**Summary:**

This paper develops a mean-field theory of dropout as a relevant perturbation of critical signal propagation at the edge of chaos. The authors derive scaling laws for the dropout induced deformation of the perfect-alignment fixed point of correlation map and show that smooth and kinked activations fall into distinct universality classes denoted by distinct critical exponents. The MFT motivates optimal dropout schedules under a fixed budget that are validated by two architectures, MLPs and ViTs.

**Compliance With Llm Reviewing Policy:**

Affirmed.

**Final Justification:**

Authors addressed my primary concerns in their rebuttal. They also provided the missing limitations section. Hence I will update my score to weak accept.

**Key Questions For Authors:**

- The scheduling argument in section 3.3 is reliant on a perturbative expansion in small h. However the experiments are conducted with mean dropout field = 0.1, which is not small.
- Also consider the weaknesses listed above

**Limitations:**

Limitations section doesn't exist.

The authors should either try to address the weaknesses or clearly mention the weaknesses in their limitation section.

**Strengths And Weaknesses:**

Strengths:

- The primary innovation in this paper is the interpretation of dropout as a symmetry breaking field in the mean field correlation dynamics. Framing dropout as a relevant perturbation in a Landau style theory unifies signal propagation, universality and regularization in a coherent framework
- The analytic development is technically strong and the theoretical estimates of critical exponents are validated through experiments
- The front-loaded dropout schedules consistently outperform constant dropout under fixed budget constraints, although the margins of improvement are rather tiny (<1% difference in test accuracies).

Weaknesses:

- The scaling exponents were solely obtained using MLPs. In order to claim that the two activations (smooth and kinked) form a universality class, the authors should consider reporting plots similar to those in Figure 1 for at least another architecture.
- Width has been kept fixed for obtaining all critical exponents. It's unclear if increasing width would keep the critical exponents the same. While the authors acknowledge in the appendix that they conduct experiments in a controlled regime (L<<N), it's unclear if the exponents are sensitive to N.
- The claim that dropout cuts off correlation length suggests a trainable depth. Existing edge-of-chaos literature emphasizes gradient propagation along with signal propagation whereas the current paper only focuses on forward correlation dynamics. It would be important to explore whether the gradient analogue of susceptibility under dropout are governed by the same exponents.

---

> ### Author Rebuttal · Authors · 2026-03-30
>
> We thank the reviewer for their thoughtful feedback. We believe two contributions beyond the Landau framing deserve emphasis in this context. First, the framework of promoting hyperparameters to depth-dependent fields and solving variational problems over them is, to our knowledge, new. We demonstrate it here for dropout scheduling, but the same machinery applies immediately to depth-dependent learning rates, residual-branch strengths, or any layer-wise hyperparameter that enters the mean-field recursion; e.g. (Yang et al., 2017) show that optimal initialization cannot be depth-independent, excluding He/Xavier as globally optimal, and our formalism provides the natural language for posing and solving such problems. We see the dropout application as the first worked example of a broader programme. Second, the fixed point is not destroyed by dropout but perturbatively displaced, so all mean-field quantities (correlation lengths, depth scales) remain well-defined, something missed in (Schoenholz et al., 2017). This resolves an open tension in the deep information propagation literature.
>
> We address the weaknesses in order. Firstly, we agree the MLP scope should be made more explicit and will do so. That said, the universality classes do extend to CNNs. The infinite-width CNN recursion is $\Sigma^{\ell+1} = A \star \mathcal{C}(\Sigma^{\ell})$ (Xiao et al., 2018), where $A$ is the channel-averaging operator induced by the convolutional filter and $\mathcal{C}$ is the same Gaussian channel as the MLP. On the translation-invariant sector the averaging operator acts trivially, collapsing the CNN map to the scalar MLP recursion. Our exponents depend only on the local analytic structure of $\bar{F}\_{\rho}$ near $c=1$ ($m^2$ vs. $m^{3/2}$), which lives entirely inside $\mathcal{C}$ and is independent of $A$. CNNs add a Fourier-mode hierarchy $\xi^{-1}\_{\alpha,\alpha'} = -\log \lvert \lambda\_{\alpha,\alpha'} \chi\_{c^\ast} \rvert$, but this dressing is multiplicative and does not alter the exponents. We could include a short discussion of this at the reviewer's request.
>
> For ResNets, (Yang et al., 2017) show that skip connections change large-depth convergence from exponential to polynomial, and the norm drifts rather than relaxing to a fixed point. However, the residual branch still passes through the same $\mathcal{C}(q,c)$, so the smooth/kinked local structure is unchanged. Indeed, (Yang et al., 2017) already observe qualitatively different tanh vs. ReLU asymptotics, consistent with our two classes, and unexplained until now. A full dropout-deformed ResNet treatment beyond the scope of this work and currently under investigation.
>
> We acknowledge that several of these extensions were omitted from the submission to not obfuscate the main message; the mathematical machinery is already dense, and we felt CNN or ResNet analyses would add little. We are happy to include concise versions of the arguments above in a revised appendix.
>
> Secondly, the exponents are properties of the $N \to \infty$ map $\bar{F}\_{\rho}(c)$; as mentioned in our draft, finite-width corrections are suppressed by $L/N$ (Roberts et al., 2022), where it is discussed that finite width corrections do not alter universality classes. These $1/N$ corrections shift $q^\ast, c^\ast$ perturbatively but do not alter the local analytic structure that determines the exponents. At $L/N \approx 0.01$ in our experiments, any width-dependent shift in the fitted exponents would be parametrically smaller than the dropout effects.
>
> Finally, the backward and forward depth scales are controlled by the same displaced fixed point and therefore the same exponents; (Schoenholz et al., 2017) showed that the backward two-input gradient covariance recurrence shares the same eigenvalue $\chi\_{c^\ast}$ as the forward map. With dropout, the duality persists: the diagonal gradient recursion acquires a factor of $1/\rho$ while the off-diagonal does not, exactly mirroring the forward asymmetry that generates $h$. Thus, the backward depth scale inherits the same displaced fixed point and Landau structure, and the exponents are identical. We will make this duality explicit in a revision.
>
> Regarding the key question on perturbative validity: at $\bar{h}=0.1$ the subleading corrections are $\mathcal{O}(h^2) \sim 10^{-2}$, an order of magnitude below the leading term. We can verify this directly from our own results: comparing the perturbative predictions against the exact (iterated) recursion in Fig. 1 and Table 3, the fitted exponents agree with the analytic values to within $3-5$% across all quantities. This value was chosen to align with typical practical dropout rates; we would be happy to include sweeps over additional values of $\bar{h}$ to demonstrate robustness.
>
>  We will also add a dedicated limitations section consolidating the limitations discussed throughout the paper and those raised in these reviews.

---

> > ### Author Rebuttal · Reviewer_uFjt · 2026-04-04
> >
> > I thank the authors for their detailed response. I appreciate the argument that universality classes are governed by the local analytic structure and this should be discussed in the main text. Please also make the duality between backward and forward depth scales more explicit. If you could also include the limitations section that you intend to add to your main text, in one of your comments, either here or to any other reviewer, that would be great!
> >
> > And please include sweeps over additional values of $\bar{h}$. I would raise my score to 4.

---

> > > ### Author Response · Authors · 2026-04-07
> > >
> > > We thank the reviewer again for the careful reading and constructive feedback. We are glad that the clarification regarding universality classes was helpful, and we will incorporate that discussion directly into the main text as suggested. We will also make the connection to backpropagation explicit, emphasizing the dual role of the correlation length in controlling both signal propagation and gradient propagation.
> > >
> > > To further strengthen the empirical case, we have added both a sweep in the mean dropout field $\bar h$ and a width sweep, both of which are available in the anonymous repository {https://anonymous.4open.science/r/reviewer-supplement-dropout-sweeps-E4D5/README.md}. These additional experiments consider widths ranging from $N = 64$ to $N = 1024$, and $0.005 \leq \bar h \leq 0.15$, and show that the gains in out of sample performance persist throughout the regime where the theory is expected to apply, only weakening near the boundaries of applicability (high dropout, or overly narrow networks). We hope these additional results make the regime of validity and the empirical robustness of our conclusions clearer and stronger.
> > >
> > > We also include below a dedicated limitations section that consolidates the main caveats discussed in the paper and raised in the review. This section is, of course, subject to any further comments that the reviewer feels should be emphasized more strongly.
> > >
> > > Limitations section:
> > >  As discussed in Section 2, mean field theory provides the leading description of MLPs at large width, with finite width corrections controlled perturbatively by the depth to width ratio $L/N$ (Roberts and Yaida, 2022). Our experiments were deliberately carried out in this regime, but the present analysis does not include these corrections explicitly. While such corrections are not expected to alter the universality class classification, they accumulate with depth and may induce subleading modifications to the optimal dropout profile. A systematic finite width scaling study would therefore strengthen the present results.
> > >
> > > A second limitation is architectural scope. Although large width mean field limits are known for CNNs and residual architectures (Xiao et al., 2018; Yang et al., 2017), we do not derive the corresponding dropout deformed Landau theories here. We expect CNNs to inherit the same local smooth versus kinked mechanism, but residual architectures are subtler, since skip connections can modify large-depth propagation and relax information bottlenecks while preserving the local universality structure. Extending the analysis to include residual branch strength as an additional control parameter, in the spirit of related near critical residual analyses (Yang et al., 2017; Noci et al., 2022), would place the Transformer results on firmer theoretical footing and would also open the door to a similar variational treatment of residual connections themselves.
> > >
> > > Thirdly, the present work focuses primarily on forward correlation dynamics. The interpretation of the correlation length $\xi_c$ as a trainable depth scale is motivated by its known dual role in gradient propagation (Schoenholz et al., 2017), but we do not explicitly derive the dropout deformed backward recursion or the associated gradient susceptibilities here. Making this duality explicit in the presence of dropout would strengthen the mechanistic interpretation of the theory. Given this duality, it would be interesting to also consider train time regularizers and study if these results can be directly transferred to training, studying how different kinds of regularization affect propagation of gradients. More broadly, it would also be interesting to understand whether other depth-dependent hyperparameters or regularization schemes, such as $L_2$ regularization, can be treated within the same framework.
> > >
> > > Finally, the theory is an initialization theory rather than a theory of full training dynamics. It characterizes signal propagation and schedule design near random initialization, but it does not model how the relevant observables evolve once feature learning substantially reshapes the representation. Extending the same framework to training time, dropout warm-up, or adaptive schedules could therefore broaden its practical scope.
> > >
> > >
> > > We hope this limitations section addresses the reviewer’s main concerns and that, together with the additional parameter sweeps, it clarifies both the scope and the robustness of our results. If the reviewer feels that these additions resolve the remaining issues, we would be very grateful for reconsideration of the score to a 4 as previously suggested.

---

### Decision · Program_Chairs · 2026-04-30

**Decision:**

Accept (regular)

**Comment:**

The reviewers trend mainly towards acceptance. The authors have addressed a number of issues raised by the reviewers. The paper gives a thorough analysis of the effect of dropout at the edge-of-chaos.